# SERENA: A Unified Stochastic Recursive Variance Reduced Gradient Framework for Riemannian Non-Convex Optimization

**Yan Liu** [* 1 2]  **Mingjie Chen** [* 3]  **Chaojie Ji** [4]  **Hao Zhang** [3]  **Ruxin Wang** [3]

## Abstract

Recently, the expansion of Variance Reduction (VR) to Riemannian stochastic non-convex optimization has attracted increasing interest. Inspired by recursive momentum, we first introduce Stochastic Recursive Variance Reduced Gradient (SRVRG) algorithm and further present Stochastic Recursive Gradient Estimator (SRGE) in Euclidean spaces, which unifies the prevailing variance reduction estimators. We then extend SRGE to Riemannian spaces, resulting in a unified Stochastic rEcursive vaRiance reducEd gradieNt frAmework (SERENA) for Riemannian non-convex optimization. This framework includes the proposed R-SRVRG, R-SVRRM, and R-Hybrid-SGD methods, as well as other existing Riemannian VR methods. Furthermore, we establish a unified theoretical analysis for Riemannian non-convex optimization under retraction and vector transport. The IFO complexity of our proposed R-SRVRG and R-SVRRM to converge to $\varepsilon$-accurate solution is $\mathcal{O}\left(\min\{n^{1/2}\varepsilon^{-2}, \varepsilon^{-3}\}\right)$ in the finite-sum setting and $\mathcal{O}\left(\varepsilon^{-3}\right)$ for the online case, both of which align with the lower IFO complexity bound. Experimental results indicate that the proposed algorithms surpass other existing Riemannian optimization methods.

## 1. Introduction

Riemannian optimization have received increasing interest in machine learning (Sato et al., 2019; Zhou et al., 2021;

---
*Equal contribution. [1]School of Statistics and Data Science, Nankai University, Tianjin, China [2]Department of Applied Mathematics, The Hong Kong Polytechnic University, Hong Kong, China [3]Shenzhen Institutes of Advanced Technology, Chinese Academy of Sciences, Shenzhen, China [4]Department of Mathematics, The University of British Columbia, Vancouver, Canada. Correspondence to: Hao Zhang <h.zhang10@siat.ac.cn>, Ruxin Wang <rx.wang@siat.ac.cn>.

*Proceedings of the 42$^{nd}$ International Conference on Machine Learning*, Vancouver, Canada. PMLR 267, 2025. Copyright 2025 by the author(s).

Han & Gao, 2021b; Dodd et al., 2024). In this paper, we focus on the following finite-sum or online optimization problems on a Riemannian manifold $\mathcal{M}$.

$$\min_{x\in\mathcal{M}} \ f(x) := \begin{cases} \dfrac{1}{n}\sum_{i=1}^{n} f_i(x) & \text{(finite-sum)}, \\ \mathbb{E}\left[f(x,\omega)\right] & \text{(online)}, \end{cases} \quad (1)$$

where $f : \mathcal{M} \to \mathbb{R}$ is a smooth non-convex function. The finite-sum case represents the minimization of the average of $n$ component functions, capturing the empirical risk minimization problem. When $n$ is very large or even infinite, $f(x)$ is typically modeled via the online form, which is indexed by a random variable $\omega$. Numerous applications can be expressed as given in (1), including principal component analysis (PCA) (Sato et al., 2019), low-rank matrix completion (LRMC) (Boumal & Absil, 2011), Riemannian centroid (RC) computation (Yuan et al., 2016), among others.

Due to the special geometric structure of the parameter space involved in such optimization problems, traditional Euclidean optimization methods may not be the optimal choice in these situations. Riemannian optimization method (Absil et al., 2008) directly advances the iterative solution along the geodesic path toward the optimum, thereby preserving the geometric structure of the problem, and effectively addressing these issues. In fact, by exploiting intrinsic properties of Riemannian manifold, the problem (1) can be regarded as an unconstrained optimization problem defined over the manifold spaces. One of the classical methods is the Riemannian steepest descent (R-SD) algorithm (Udriste, 2013; Zhang & Sra, 2016). However, the requirement for each iteration of R-SD to traverse $n$ component functions renders the process impractical when $n$ is extremely large. In contrast, the Riemannian stochastic gradient descent (R-SGD) (Bonnabel, 2013) method computes the stochastic gradient for each iteration of one sample (or a mini-batch samples), significantly reducing the computational cost per iteration. However, R-SGD exhibits slow convergence attributed to the high variance (Bottou et al., 2018).

Variance reduction (Gower et al., 2020) has also been extended to Riemannian manifolds for improving performance. For instance, Riemannian stochastic variance reduced gradient (R-SVRG) (Zhang et al., 2016; Kasai et al., 2016;

Sato et al., 2019), Riemannian stochastic recursive gradient (R-SRG) (Kasai et al., 2018b; Han & Gao, 2021b), and Riemannian stochastic path integrated differential estimator (R-SPIDER) (Zhang et al., 2018; Zhou et al., 2021) are generalized from the Euclidean versions in (Johnson & Zhang, 2013; Nguyen et al., 2017; Fang et al., 2018). Variance reduction enjoys more favorable complexity in Riemannian non-convex optimization. Specifically, the Incremental First-order Oracle (IFO) complexity of R-SVRG to achieve an $\varepsilon$-accurate solution is $\mathcal{O}\left(n^{2/3}\varepsilon^{-2}\right)$ (Han & Gao, 2021b), which improves upon $\mathcal{O}\left(\varepsilon^{-4}\right)$ of R-SGD. The R-SRG and R-SPIDER algorithms further enhance the complexity to $\mathcal{O}\left(\varepsilon^{-3}\right)$ (Zhou et al., 2021), matching the lower bound for stochastic non-convex optimization complexity in Euclidean spaces (Arjevani et al., 2023). However, a "hybrid" algorithm that combines two existing stochastic estimators through a convex combination to design a hybrid offspring—inheriting the advantages of its underlying estimators—has yet to be developed in the field of Riemannian stochastic optimization.

In the Euclidean search space, Tran-Dinh et al. (2022) designed a hybrid stochastic estimator to balance the variance and bias, which can be regarded as a convex combination of the SARAH-type estimator and the SGD estimator. Based on this, they proposed the Hybrid-SGD algorithm, which can be seen as an extension of the stochastic recursive momentum (STROM) method. Although they stated that their approach could be extended to cover SVRG-type methods, they did not provide explicit formulations. A general recursive momentum estimator, stochastic variance reduced recursive momentum (SVRRM) (Liao et al., 2024), was proposed by incorporating loopless-SVRG (Kovalev et al., 2020) into STORM. Jiang et al. (2024) also introduced a similar stochastic estimator, but these approaches iterate based on the Euclidean space.

While integrating SVRG-type estimators into hybrid stochastic estimator allows us to achieve a unified form of variance reduction algorithms, there is currently no related work on Riemannian stochastic optimization. Furthermore, the theoretical analyses of various Riemannian stochastic algorithms differ significantly, and in some cases, the construction of a Lyapunov function is required. A natural question arises: "Is there a unified formulation for Riemannian stochastic variance reduction methods and, consequently, a simple and unified theoretical framework?" This paper provides a compelling affirmative answer. Specifically, our main contributions are summarized as follows.

- Motivated by recursive momentum, we propose the SRVRG estimator and extend it to Riemannian manifolds. We establish an improved complexity bound for Riemannian SRVRG (R-SRVRG), which achieves the optimal complexity.

- We also develop the R-SVRRM and R-Hybrid-SGD algorithms by extending the SVRRM and Hybrid-SGD estimators to Riemannian manifolds.
- We introduce the stochastic recursive gradient estimator, which unifies variance reduction methods (see Table 1). We further propose the Stochastic rEcursive vaRiance rEduced gradieNt frAmework (SERENA) for Riemannian optimization by extending SRGE to Riemannian spaces. Our proposed algorithms can be regarded as special cases of this framework (see Section 5).
- By providing an upper bound on the variance of Riemannian stochastic estimator within the SERENA framework, we establish a unified theoretical analysis for general non-convex functions under retraction and vector transport. Specifically, we derive convergence results and IFO complexity for several algorithms under both finite-sum and online settings.
- The experimental results on various tasks and datasets demonstrate that our proposed R-SRVRG and R-SVRRM algorithms outperform existing methods.

## 2. Preliminaries

Riemannian manifold (Absil et al., 2008; Boumal, 2023) $\mathcal{M}$ is a manifold that is equipped with a smoothly varying inner product $\langle \cdot, \cdot \rangle_x$ on tangent space $\mathrm{T}_x\mathcal{M}$ for every $x \in \mathcal{M}$. The induced norm is given by $\|u\|_x := \sqrt{\langle u, u \rangle_x}$ for $u \in \mathrm{T}_x\mathcal{M}$. In iterative optimization algorithms on manifold $\mathcal{M}$, an iteration is performed by following geodesics emanating from $x$ and tangent to $u \in \mathrm{T}_x\mathcal{M}$. A locally shortest path on the manifold with constant speed is called a geodesic curve $\gamma : [0, 1] \to M$, which is a generalized concept of straight lines in Euclidean space.

If $\mathcal{M}$ is a complete manifold, the exponential mapping is defined for all vectors $u \in \mathrm{T}_x\mathcal{M}$ (Absil et al., 2008; Sato et al., 2019), then we can use the exponential mapping to update. However, for some Riemannian manifolds, the closed form of the exponential map is not available. Alternatively, we can retract the variable $x$ into the manifold $\mathcal{M}$ by defining retraction to update. Retraction $R_x : \mathrm{T}_x\mathcal{M} \to \mathcal{M}$ approximates the exponential map and maps a tangent vector $\xi$ to $z = R_x(\xi)$ such that $R_x(0) = x$ and $\mathrm{D}R_x(0)[\xi] = \xi$. The retraction curve is defined as $c(t) := R_x(t\xi)$ and $R_x^{-1} : \mathcal{M} \to \mathrm{T}_x\mathcal{M}$ denotes the inverse retraction if $R$ has smooth bijection. One advantage of using retraction is that it can reduce computational costs compared to the exponential map.

A common approach for performing the addition of tangent vectors in different tangent spaces is to use the parallel transport $P_x^y : \mathrm{T}_x\mathcal{M} \to \mathrm{T}_y\mathcal{M}$, which transports vector on the geodesic curve $\gamma$ that connects $x$ to $y$ such that the induced vector fields are in parallel. However, the parallel

translation can sometimes be computationally expensive, and it does not have a explicit formula for certain manifolds (Sato et al., 2019). Vector transport $\mathcal{T}_x^z : \mathrm{T}\mathcal{M} \oplus \mathrm{T}\mathcal{M} \to \mathrm{T}\mathcal{M}$ is used as an alternative (Absil et al., 2008), it satisfies: 1) $\mathcal{T}$ has an associated retraction $R$, i.e., for $x \in \mathcal{M}$ and $\iota, \xi \in \mathrm{T}_x\mathcal{M}, \mathcal{T}_\xi(\iota)$ is a tangent vector at $R_x(\iota)$; 2) $\mathcal{T}_{0_x}(\xi) = \xi$, where $\xi \in \mathrm{T}_x\mathcal{M}$ and $x \in \mathcal{M}$; 3) $\mathcal{T}_\xi(ay_1 + by_2) = a\mathcal{T}_\xi(y_1) + b\mathcal{T}_\xi(y_2)$, where $a, b \in \mathbb{R}, \xi, y_1, y_2 \in \mathrm{T}_x\mathcal{M}$, and $x \in \mathcal{M}$. Similar to (Zhou et al., 2021; Han & Gao, 2021a;b), we implicitly assume vector transport is isometric, which means that $\langle \mathcal{T}_x^z u, \mathcal{T}_x^z v \rangle_z = \langle u, v \rangle_x$ for all $u, v \in \mathrm{T}_x\mathcal{M}$ and $x, z \in \mathcal{M}$.

In this paper, our analysis focuses on retraction and vector transport, which are more general and efficient. It is common to use the incremental first-order oracle (IFO) complexity to measure the total complexity of stochastic optimization algorithms in achieving $\varepsilon$-accurate solution.

**Definition 2.1.** ($\epsilon$-accurate solution and IFO complexity) The $\epsilon$-accurate solution of a stochastic algorithm is an output $x$ for which the expected gradient norm does not exceed $\varepsilon$. i.e., $\mathbb{E}\left[\|\operatorname{grad} f(x)\|\right] \leqslant \varepsilon$. For problem (1), an Incremental first-order oracle (IFO) takes an index $i \in \{1, \dots, n\}$ and a point $x$, and returns the pair $(f_i(x), \operatorname{grad} f_i(x) \in \mathrm{T}_x\mathcal{M})$.

The IFO complexity effectively captures the overall computational cost of a first-order Riemannian algorithm, as the evaluations of the objective function and gradient typically dominate the per-iteration computations.

### 2.1. Assumptions

We present the following assumptions, which are necessary for the convergence analysis. Note that these assumptions are standard in the analysis of optimization algorithms involving retraction and vector transport (Kasai et al., 2018b; Sato et al., 2019; Han & Gao, 2021b; Zhou et al., 2021).

**Assumption 2.2.** (1) Iterate sequences generated by algorithms stay continuously in a neighbourhood $\mathcal{X} \subset \mathcal{M}$ around an optimal solution $x_*$. Additionally, $\mathcal{X}$ is a totally retractive neighbourhood of $x_*$ where retraction $R$ is a diffeomorphism.

(2) Assume each loss $f_i$ and the objective function $f$ are twice continuously differentiable. Norms of Riemannian gradient. That is $\|\operatorname{grad} f_i(x)\| \leqslant G$ for all $x \in \mathcal{X}$, where $G > 0$.

(3) Stochastic gradient $\operatorname{grad} f_i(x)$ is unbiased and has bounded variance. That is, for all $x \in \mathcal{X}$, $\mathbb{E}\left[\operatorname{grad} f_i(x)\right] = \operatorname{grad} f(x)$, $\mathbb{E}\left[\|\operatorname{grad} f_i(x) - \operatorname{grad} f(x)\|^2\right] \leqslant \sigma^2$.

(4) The objective function $f$ is retraction $L$-smooth with respect to retraction $R$. That is, for all $x, y = R_x(\xi) \in \mathcal{X}$, there exists a constant $L > 0$ such that $f(y) \leqslant f(x) + \langle \operatorname{grad} f(x), \xi \rangle + \frac{L}{2}\|\xi\|^2$.

(5) The function $f$ is average retraction $L_l$-Lipschitz. That is, there exists a constant $L_l > 0$ such that for all $x, y \in \mathcal{X}$, $\mathbb{E}\left[\|\operatorname{grad} f_i(x) - P_y^x \operatorname{grad} f_i(y)\|\right] \leqslant L_l\|\xi\|$, where $P_y^x$ is the parallel transport operator.

(6) Difference between vector transport $\mathcal{T}$ and parallel transport $P$ associated with the same retraction $R$ is bounded. That is, for all $x, y = R_x(\xi) \in \mathcal{X}$ and $u \in \mathrm{T}_x\mathcal{M}$, there exists a constant $\theta \geq 0$, such that $\|\mathcal{T}_x^y u - P_x^y u\| \leqslant \theta\|\xi\|\|u\|$.

The following assumptions are also standard (Sato et al., 2019; Zhou et al., 2021; Han & Gao, 2021b).

**Assumption 2.3.** (1) The neighbourhood $\mathcal{X}$ is also a totally normal neighbourhood of $x^*$ where exponential map is a diffeomorphism.

(2) There exists $\mu, \nu, \delta_{\mu,\nu} > 0$ where for all $x, y = R_x(\xi) \in \mathcal{X}$ with $\|\xi\| \leqslant \delta_{\mu,v}$, we have $\|\xi\| \leqslant \mu d(x, y)$ and $d(x, y) \leqslant \nu\|\xi\|$.

*Notations.* For notation simplicity, we omit the subscripts for norm and inner product. Define $\Delta_k^s = v_k^s - \operatorname{grad} f(x_k^s)$, $\Delta_k = v_k - \operatorname{grad} f(x_k)$, and $\tilde{\Delta}_0 = \mathbb{E}\left[f(\tilde{x}_0)\right] - f(x_*)$. Denote $[n] = \{1, \dots, n\}$, $\tilde{L} = L_l + \theta G$. $\nabla f_i(x)$ and $\operatorname{grad} f_i(x)$ represent stochastic gradient in Euclidean space and stochastic Riemannian gradient of $f_i(x)$, respectively. It holds that $\mathbb{E}\left[\operatorname{grad} f_i(x)\right] = \operatorname{grad} f(x)$. $\operatorname{grad} f_{\mathcal{B}}(x) = \frac{1}{|\mathcal{B}|}\sum_{i \in \mathcal{B}} \operatorname{grad} f_i(x)$ is a mini-batch Riemannian stochastic gradient on $\mathrm{T}_x\mathcal{M}$. We use $f(n) = \mathcal{O}(g(n))$ to represent the existence of constants $c$ and $N$, such that $|f(n)| \leqslant c|g(n)|$ always holds for all $n \geq N$. We denote $\tilde{\mathcal{O}}(\cdot)$ to further hide poly-logarithmic factors and use $f(n) = \Theta(g(n))$ to represent the existence of $c_1, c_2$, and $N$, such that $c_1|g(n)| \leqslant |f(n)| \leqslant c_2|g(n)|$ holds for all $n \geq N$.

## 3. Stochastic Recursive Variance Reduced Gradient Algorithm

Recursive momentum, as proposed by Cutkosky & Orabona (2019), serves to achieve variance reduction and can essentially be regarded as a specific case of the Hybrid-SGD (Tran-Dinh et al., 2022) algorithm. Hybrid-SGD is derived from the combination of SARAH-type estimator and SGD.

$$
\begin{aligned}
v_k &= \overbrace{(1 - \beta)\left(v_{k-1} - \nabla f_{i_k}(x_{k-1})\right) + \nabla f_{i_k}(x_k)}^{\text{STORM}} \\
&\approx \overbrace{(1-\beta)\underbrace{\left(v_{k-1} + \nabla f_{i_k}(x_k) - \nabla f_{i_k}(x_{k-1})\right)}_{\text{SARAH-type estimator}} + \beta\nabla f_{j_k}(x_k)}^{\text{Hybrid - SGD}}.
\end{aligned}
$$

The first row represents the STORM estimator, while the second row corresponds to the Hybrid-SGD. When $i_k = j_k$, then Hybrid-SGD reduce to STORM estimator. We propose a Riemannian extension of Hybrid-SGD, called R-Hybrid-

SGD, as outlined below,

$$v_k = (1 - \beta)\mathcal{T}_{x_{k-1}}^{x_k}(v_{k-1} - \text{grad } f_{j_k}(x_{k-1}))$$
$$+ \beta \text{ grad } f_{i_k}(x_k) + (1 - \beta) \text{ grad } f_{j_k}(x_k). \quad (2)$$

When $i_k = j_k$, we get R-SRM (Han & Gao, 2021a). Inspired by hybrid estimator, we propose stochastic recursive variance reduced gradient (SRVRG) algorithm, which integrates the SVRG-type estimator with SARAH. In the $s$-th outer loop,

$$v_k^s = (1-\beta)v_{k-1}^s + \beta u_k^s + (1-\beta)\left(\nabla f_{i_k}(x_k^s) - \nabla f_{i_k}(x_{k-1}^s)\right), \quad (3)$$

where $u_k^s = \nabla f_{j_k}(x_k^s) - \nabla f_{j_k}(\tilde{x}_{s-1}) + \nabla f(\tilde{x}_{s-1})$. $\tilde{x}_{s-1}$ represents the point at which the true gradient is calculated in the outer loop of SVRG algorithm. Note that if we choose $i_k = j_k$, SRVRG reduces to SVRRM (Liao et al., 2024) or SSVR-FS (Jiang et al., 2024) algorithm. The primary advantage of SRVRG lies in its ability to select a larger parameter $\beta$ when $u_k^s$ is an SVRG-type estimator, which facilitates a more rapid reduction in the variance of $v_k^s$. By extending it to Riemannian manifolds, we obtain Riemannian SRVRG (R-SRVRG). The key step of R-SRVRG is similar to (3) except the vector transport $\mathcal{T}$.

$$v_k^s = (1 - \beta)\mathcal{T}_{x_{k-1}^s}^{x_k^s}\left(v_{k-1}^s - \text{grad } f_{\mathcal{I}_k}(x_{k-1}^s)\right)$$
$$+ \beta\left(\text{grad } f_{\mathcal{J}_k}(x_k^s) - \mathcal{T}_{x_0^s}^{x_k^s}\left(\text{grad } f_{\mathcal{J}_k}(x_0^s) - v_0^s\right)\right) \quad (4)$$
$$+ (1 - \beta) \text{ grad } f_{\mathcal{I}_k}(x_k^s),$$

where $x_0^s = \tilde{x}_{s-1}$, $v_0 = \text{grad } f_{\mathcal{B}}(x_0^s)$ and $|\mathcal{B}| = b$, $|\mathcal{I}| = |\mathcal{J}| = b'$. Our theoretical results (Theorem 5.3 and Corollary 5.4) show that the IFO complexity of R-SRVRG matches the lower-bound complexity (Zhou & Gu, 2019; Arjevani et al., 2023). Additionally, experimental results indicate that R-SRVRG significantly outperforms R-Hybrid-SGD. Setting $\mathcal{I}_k = \mathcal{J}_k$ enables the derivation of the Riemannian SVRRM (R-SVRRM) algorithm,

$$v_k^s = (1 - \beta)\mathcal{T}_{x_{k-1}^s}^{x_k^s}\left(v_{k-1}^s - \text{grad } f_{\mathcal{I}_k}(x_{k-1}^s)\right)$$
$$+ \beta\left(\text{grad } f_{\mathcal{I}_k}(x_k^s) - \mathcal{T}_{x_0^s}^{x_k^s}\left(\text{grad } f_{\mathcal{I}_k}(x_0^s) - v_0^s\right)\right) \quad (5)$$
$$+ (1 - \beta) \text{ grad } f_{\mathcal{I}_k}(x_k^s).$$

The specific algorithms are provided in the Appendix A.

# 4. Stochastic Recursive Variance Reduced Gradient Framework for Riemannian Non-convex Optimization

In this section, we first introduce the Stochastic Recursive Gradient Estimator (SRGE) and then propose a Stochastic Recursive Variance Reduced Gradient framework (SERENA) for Riemannian non-convex optimization, which ex-

tends SRGE to the Riemannian spaces. Notably, our framework also encompasses several existing Riemannian optimization methods, such as R-SGD, R-SVRG (Zhang et al., 2016; Sato et al., 2019), and R-SRG (Kasai et al., 2018b).

## 4.1. Stochastic Recursive Gradient Estimator

Motivated by hybrid stochastic estimator (Tran-Dinh et al., 2022), we introduce the stochastic recursive gradient estimator (SRGE) as follows,

$$v_k = (1 - \beta)v_{k-1} + \beta u_k + (1 - \beta)(w_k - w_{k-1}), \quad (6)$$

where $\mathbb{E}[u_k] = \nabla f(x_k)$, $\mathbb{E}[w_k - w_{k-1}] = \nabla f(x_k) - \nabla f(x_{k-1})$. Let us focus on the "error in $v_k$ of (6)" which we denote as $\Delta_k = v_k - \nabla f(x_k)$. It is easy to see that

$$\mathbb{E}[\|\Delta_k\|] \leqslant (1 - \beta)\mathbb{E}[\|\Delta_{k-1}\|] + \beta\mathbb{E}[\|u_k - \nabla f(x_k)\|]$$
$$+ (1 - \beta)\mathbb{E}[\|(w_k - w_{k-1} - (\nabla f(x_k) - \nabla f(x_{k-1})))\|].$$

The second term on the right-hand side is expected to be controlled by a sufficiently small $\beta$. Under the bounded variance assumption (Assumption 2.2 (3)), it is possible to establish a non-increasing upper bound for $\mathbb{E}[\|u_k - \nabla f(x_k)\|^2]$. For instance, if $u_k$ is a variance reduced estimator, then $\mathbb{E}[\|u_k - \nabla f(x_k)\|^2] \to 0$ when $k \to \infty$ (Gower et al., 2020). The third term is of order $\mathcal{O}(\|x_k - x_{k-1}\|)$ under the assumption of $L$-smooth, which can be controlled by a small step size $\eta$. Consequently, the variance is expected to decrease. On the other hand, the bias of $v_k$ is smaller than $v_{k-1}$ when $\beta < 1$ as $\text{Bias}[v_k] = \|\mathbb{E}[v_k - \nabla f(x_k)]\| < (1 - \beta)\|v_{k-1} - \nabla f(x_{k-1})\|$. Therefore, we can get improved performance by balancing bias and variance of SRGE. Indeed, if $\beta = 0$ is chosen in (6) and $v_0$ repre-

*Table 1.* Stochastic Recursive Gradient Estimator. Given $\beta$, the second and third columns represent the types of stochastic gradient estimators for $u$ and $w$, respectively, while the last column indicates the corresponding algorithms.

| $\beta$ | $u_k$ | $w_k, w_{k-1}$ | Methods |
|---|---|---|---|
| $\beta = 0$ | — | SGD | SARAH-type |
| $\beta = 1$ | SGD | — | SGD |
| | SVRG-type | — | SVRG-type |
| $\beta \in (0, 1)$ | SGD | SGD | STORM |
| | SGD | SARAH | Hybrid-SGD |
| | SVRG | SGD | SVRRM |
| | SVRG | SARAH | **SRVRG** |

sents the true gradient, then SRGE reduce to SARAH-type estimator. When $\beta = 1$, SRGE can specifically configure $u_k$ to correspond to either SGD or SVRG-type estimator. Furthermore, if $\beta \in (0, 1)$, SRGE can cover several other algorithms, including Hybrid-SGD, STORM, SVRRM, and our proposed SRVRG, see Table 1 for details.

## 4.2. Riemannian Stochastic Recursive Variance Reduced Gradient Methods

By extending SRGE to Rimannian manifold $\mathcal{M}$, we establish a unified stochastic recursive variance reduced gradient framework (SERENA) for Riemannian stochastic algorithms as presented in Algorithm 1. In the $s$-th outer loop,

$$v_k^s = (1 - \beta)\,\mathcal{T}_{x_{k-1}^s}^{x_k^s}\left(v_{k-1}^s - w_{k-1}^s\right) + \beta u_k^s + (1 - \beta)\,w_k^s, \quad (7)$$

when $u_k^s$ is SVRG-type estimator, $u_k^s = \mathrm{grad}\,f_{\mathcal{I}_k}(x_k^s) - \mathcal{T}_{x_0^s}^{x_k^s}(\mathrm{grad}\,f_{\mathcal{I}_k}(x_0^s) - v_0^s)$, when $u_k^s$ is SG estimator, $u_k^s = \mathrm{grad}\,f_{\mathcal{I}_k}(x_k^s)$. Vector transport is employed to integrate gradient information in (7), as the Riemannian stochastic gradient of $x_k^s$, $x_{k-1}^s$, and $x_0^s$ are defined on disjoint tangent spaces. The SERENA framework includes various Riemannian stochastic optimization algorithms, among which are the algorithms we proposed. Several of these are listed below. For single-loop algorithms, such as R-Hybrid-SGD and R-SRM, as well as loopless algorithms like R-PAGE, $S = 1$ in Algorithm 1. And we omit the superscript for these algorithms to simplify notation, as in Equations (8) and (10).

---

**Algorithm 1** Stochastic rEcursive vaRiance reducEd gradiEnt frAmework (SERENA)

---

**Input:** Step size $\eta$, outer loop size $S$, inner loop size $m$, batch size $b$, $b'$, initial point $\tilde{x}_0$.

1: **for** $s = 1, 2, \ldots, S$ **do**
2: $\quad x_0^s = \tilde{x}_{s-1}$.
3: $\quad$ Sample $\mathcal{B}$ uniformly at random from $[n]$ of size $b$.
4: $\quad v_0^s = \mathrm{grad}\,f_{\mathcal{B}}(x_0^s)$.
5: $\quad x_1^s = R_{x_0^s}(-\eta v_0^s)$.
6: $\quad$ **for** $k = 1, \ldots, m - 1$ **do**
7: $\qquad$ Sample $\mathcal{I}_k$ and $\mathcal{J}_k$ uniformly at random from $[n]$ with $|\mathcal{I}_k| = |\mathcal{J}_k| = b'$.
8: $\qquad$ Calculate SERENA estimator $v_k^s$.
9: $\qquad x_{k+1}^s = R_{x_k^s}(-\eta v_k^s)$.
10: $\quad$ **end for**
11: $\quad \tilde{x}_s = x_m^s$
12: **end for**
13: **Output:** $x_\zeta$ uniformly selected at random from $\{x_k^s\}$, $k \in [m-1], s \in [S]$.

---

- R-SRM (Han & Gao, 2021a) was proposed by extending STORM to Rimannian manifold.

$$\begin{aligned} v_k = (1 - \beta)\mathcal{T}_{x_{k-1}}^{x_k}\left(v_{k-1} - \mathrm{grad}\,f_{\mathcal{I}_k}(x_{k-1})\right) \\ + \mathrm{grad}\,f_{\mathcal{I}_k}(x_k). \end{aligned} \quad (8)$$

- Riemannian SVRG (R-SVRG) (Zhang et al., 2016; Kasai et al., 2016; Han & Gao, 2021b) is captured

by setting $\beta = 1$ and choosing SVRG-type estimator in (7). Furthermore, we can achieve Riemannian AbaSVRG (Han & Gao, 2021b) by implementing batch size adaptation in the outer loop.

$$v_k^s = \mathrm{grad}\,f_{\mathcal{I}_k}(x_k^s) - \mathcal{T}_{x_0^s}^{x_k^s}\left(\mathrm{grad}\,f_{\mathcal{I}_k}(x_0^s) - v_0^s\right), \quad (9)$$

where $|\mathcal{I}| = b'$.

- Riemannian probabilistic gradient estimator (R-PAGE) was proposed by Demidovich et al. (2024) as an extension of the PAGE (Li et al., 2021) in Euclidean space, where PAGE is a SARAH-type algorithm.

$$v_k = \begin{cases} \mathrm{grad}\,f_{\mathcal{B}}(x_k), & \text{with probability } p \\ \mathrm{grad}\,f_{\mathcal{I}}(x_k) + \mathcal{T}_{x_{k-1}}^{x_k}(v_{k-1} - \mathrm{grad}\,f_{\mathcal{I}}(x_{k-1})), & \text{o} \end{cases} \quad (10)$$

where $|\mathcal{B}| = b$, $|\mathcal{I}| = b'$ and "o" represents "otherwise".

## 5. Convergence Analysis for SERENA

We introduce a unified convergence theorem in this section. All proofs can be found in the Appendix B. To simplify the notation, we denote $\mathcal{L} = \tilde{L}^2 \mu^2 \nu^2$ in this section.

**Theorem 5.1.** *Suppose that Assumptions 2.2 and 2.3 hold, Let $K = Sm$ denote the number of total iterations, if there exist $\mathcal{M}_1, \mathcal{M}_2, \gamma_{k,i}, \lambda_k > 0$ such that the Riemannian stochastic estimator $v_k^s$ (7) in SERENA satisfies*

$$\mathbb{E}\left[\|\Delta_k^s\|^2\right] \leqslant \mathcal{M}_1 \eta^2 \sum_{i=0}^{k-1} \gamma_{k,i}\mathbb{E}\left[\|v_i^s\|^2\right] + \mathcal{M}_2 \lambda_k \sigma^2. \quad (11)$$

*Then we have $\mathbb{E}\left[\|\mathrm{grad}\,f(x_\zeta)\|^2\right] \leqslant \frac{\varphi\tilde{\Delta}_0}{\phi K} + \frac{\varphi\eta\mathcal{M}_2\hat{\Lambda}\sigma^2}{2\phi K} + \frac{2\mathcal{M}_2\hat{\Lambda}\sigma^2}{K}$, where $\varphi = 2\left(\mathcal{M}_1\eta^2\hat{\Gamma} + 1\right)$, $\phi = \eta\left(\frac{1 - \eta L - \eta^2\mathcal{M}_1\hat{\Gamma}}{2}\right)$, $\hat{\Gamma} = \max\{\Gamma_k^s\}$ for all $k \in [m-1], s \in [S]$, $\Gamma_k^s = \sum_{k=0}^{m-1} \gamma_{k,0}$, and $\hat{\Lambda} = \sum_{s=1}^{S}\sum_{k=0}^{m-1} \lambda_k$.*

Theorem 5.1 indicates that convergence results can be obtained when the variance of (7) has an upper bound, specifically when it satisfies (11). Next, we will present convergence results for several specific algorithms.

### 5.1. Convergence for R-SRVRG and R-SVRRM

We first prove that the variance of estimator (4) and (5) satisfy the inequality (11).

**Lemma 5.2.** *Suppose that Assumptions 2.2 and 2.3 hold. Then the variance of R-SRVRG estimator (4) has an upper bound as (11) with $\mathcal{M}_1 = \frac{\tilde{L}^2 + \beta^2\mathcal{L}m^2}{b'}$, $\mathcal{M}_2 = \frac{\mathbb{1}_{\{b<n\}}}{b}$, $\gamma_{k,i} = (1 - \beta)^{2(k-i-1)}$, and $\lambda_k = (1 - \beta)^{2(k-1)} + \beta$.*

| Stochastic Method | Riemannian Stochastic Method | Finite-sum | Online |
|---|---|---|---|
| SGD | Riemannian SGD (Hosseini & Sra, 2020) | $\mathcal{O}\left(\varepsilon^{-4}\right)$ | $\mathcal{O}\left(\varepsilon^{-4}\right)$ |
| SVRG-type | Riemannian SVRG (Han & Gao, 2021b) | $\mathcal{O}\left(n + n^{2/3}\varepsilon^{-2}\right)$ | $\mathcal{O}\left(\varepsilon^{-10/3}\right)$ |
| | Riemannian AbaSVRG (Han & Gao, 2021b) | $\mathcal{O}\left(n^{2/3}\varepsilon^{-2}\right)$ | $\mathcal{O}\left(\varepsilon^{-10/3}\right)$ |
| | **Riemannian SVRG** | $\mathbf{\mathcal{O}(n^{2/3}\varepsilon^{-2})}$ | $\mathbf{\mathcal{O}(\varepsilon^{-10/3})}$ |
| SARAH-type | Riemannian SRG (Kasai et al., 2018b) | $\mathcal{O}\left(n + \varepsilon^{-4}\right)$ | — |
| | Riemannian AbaSRG (Han & Gao, 2021b) | $\mathcal{O}\left(n^{1/2}\varepsilon^{-2}\right)$ | $\mathcal{O}\left(\varepsilon^{-3}\right)$ |
| | Riemannian SPIDER (Zhou et al., 2021) | $\mathcal{O}\left(\min\{n + n^{1/2}\varepsilon^{-2}, \varepsilon^{-3}\}\right)$ | $\mathcal{O}\left(\varepsilon^{-3}\right)$ |
| | Riemannian PAGE (Demidovich et al., 2024) | — | — |
| | **Riemannian PAGE** | $\mathbf{\mathcal{O}\left(n + n^{1/2}\varepsilon^{-2}\right)}$ | $\mathbf{\mathcal{O}\left(\varepsilon^{-3}\right)}$ |
| STORM | Riemannian SRM (Han & Gao, 2021a) | — | $\tilde{\mathcal{O}}\left(\varepsilon^{-3}\right)$ |
| | **Riemannian SRM** | $\mathbf{\mathcal{O}\left(\min\{n, \varepsilon^{-1}\} + \varepsilon^{-3}\right)}$ | $\mathbf{\mathcal{O}\left(\varepsilon^{-3}\right)}$ |
| Hybrid-SGD | **Riemannian Hybrid-SGD\*** | $\mathbf{\mathcal{O}\left(\min\{n, \varepsilon^{-1}\} + \varepsilon^{-3}\right)}$ | $\mathbf{\mathcal{O}\left(\varepsilon^{-3}\right)}$ |
| SVRRM | **Riemannian SVRRM\*** | $\mathbf{\mathcal{O}\left(\min\{n^{1/2}\varepsilon^{-2}, \varepsilon^{-3}\}\right)}$ | $\mathbf{\mathcal{O}\left(\varepsilon^{-3}\right)}$ |
| SRVRG | **Riemannian SRVRG\*** | $\mathbf{\mathcal{O}\left(\min\{n^{1/2}\varepsilon^{-2}, \varepsilon^{-3}\}\right)}$ | $\mathbf{\mathcal{O}\left(\varepsilon^{-3}\right)}$ |

*Table 2.* The IFO complexity of Riemannian stochastic methods for non-convex optimization with retraction and vector transport. The results presented in this paper are highlighted in **bold**.

*The variance of the estimator for R-SVRRM has a similar upper bound with* $\mathcal{M}_1 = \frac{2\tilde{L}^2 + 2\beta^2 \mathcal{L}m^2}{b'}$, $\mathcal{M}_2 = \frac{\mathbb{1}_{\{b<n\}}}{b}$, $\gamma_{k,i} = (1-\beta)^{2(k-i-1)}$, *and* $\lambda_k = (1-\beta)^{2(k-1)} + 2\beta$.

Consequently, we have the following convergence results for R-SRVRG and R-SVRRM.

**Theorem 5.3.** *Suppose that Assumptions 2.2 and 2.3 hold. Then the output* $x_\zeta$ *after running* $K = Sm$ *iterations of R-SRVRG algorithm satisfies* $\mathbb{E}\left[\|\mathrm{grad}\, f(x_\zeta)\|^2\right] \leqslant \frac{\varphi_1 \tilde{\Delta}_0}{\phi_1 K} + \frac{(\varphi_1 \eta + 4\phi_1)\mathbb{1}_{\{b<n\}}\sigma^2}{2\phi_1 Kb}\left(\frac{1}{\beta} + K\beta\right)$, *where* $\varphi_1 = \frac{2\eta^2\left(\tilde{L}^2 + \beta^2 \mathcal{L}m^2\right)}{\beta b'} + 2$, $\phi_1 = \frac{\eta(1-\eta L)}{2} - \frac{\eta^3\left(\tilde{L}^2 + \beta^2 \mathcal{L}m^2\right)}{2\beta b'}$. *Similarly, the output* $x_\zeta$ *of R-SVRRM algorithm satisfies* $\mathbb{E}\left[\|\mathrm{grad}\, f(x_\zeta)\|^2\right] \leqslant \frac{\varphi_2 \tilde{\Delta}_0}{\phi_2 K} + \frac{(\varphi_2 \eta + 4\phi_2)\mathbb{1}_{\{b<n\}}\sigma^2}{2\phi_2 Kb}\left(\frac{1}{\beta} + 2K\beta\right)$, *where* $\varphi_2 = \frac{4\eta^2\left(\tilde{L}^2 + \beta^2 \mathcal{L}m^2\right)}{\beta b'} + 2$, $\phi_2 = \frac{\eta(1-\eta L)}{2} - \frac{2\eta^3\left(\tilde{L}^2 + \beta^2 \mathcal{L}m^2\right)}{2\beta b'}$.

**Corollary 5.4.** *Suppose that Assumptions 2.2 and 2.3 hold. Set* $\beta = m^{-1}$ *for both R-SRVRG and R-SVRRM algorithms. Denote* $\eta = \min\{\frac{1}{2L}, \frac{\sqrt{b'}}{2\tilde{L}\sqrt{m}\sqrt{1+\mu^2\nu^2}}\}$ *for R-SRVRG, and* $\eta = \min\{\frac{1}{2L}, \frac{\sqrt{b'}}{4\tilde{L}\sqrt{m}\sqrt{1+\mu^2\nu^2}}\}$ *for R-SVRRM. Choose* $b = \min\{n, \sigma^2\varepsilon^{-2}\}$ *and* $\sqrt{\frac{\beta}{b'}} = \max\{\frac{1}{\sqrt{n}}, \varepsilon\}$

*for both R-SRVRG and R-SVRRM in finite-sum case. While under online setting, we set* $b = \Theta\left(\varepsilon^{-2}\right)$, $\sqrt{\frac{\beta}{b'}} = \varepsilon$. *Then*

$$S = \begin{cases} \Theta\left(\max\{n^{-1/2}\varepsilon^{-2}, \varepsilon^{-1}\}\right), & \text{(finite - sum)} \\ \Theta\left(\varepsilon^{-1}\right), & \text{online} \end{cases}$$

*The IFO complexity of Riemannian SRVRG or Riemannian SVRRM to obtain* $\varepsilon$*-accurate solution is*

$$\begin{cases} \mathcal{O}\left(\min\{n^{1/2}\varepsilon^{-2}, \varepsilon^{-3}\}\right), & \text{(finite-sum)} \\ \mathcal{O}\left(\varepsilon^{-3}\right), & \text{(online)} \end{cases}$$

*Remark* 5.5. We establish an $\mathcal{O}\left(\min\{n^{1/2}\varepsilon^{-2}, \varepsilon^{-3}\}\right)$ IFO complexity for non-convex finite-sum problems, which is not worse than $\mathcal{O}\left(n^{1/2}\varepsilon^{-2}\right)$. Notably, our complexity is superior in the regime $n > \mathcal{O}(\varepsilon^{-2})$, which is significant for large-scale machine learning applications. In the online setting, our result achieves the best-known complexity of $\mathcal{O}\left(\varepsilon^{-3}\right)$ (Li et al., 2021; Arjevani et al., 2023).

*Remark* 5.6. Corollary 5.4 indicates that if $\beta$ and $b'$ satisfy $\sqrt{\frac{\beta}{b'}} = \max\left\{\frac{1}{\sqrt{n}}, \varepsilon\right\}$ (finite-sum case) or $\sqrt{\frac{\beta}{b'}} = \varepsilon$ (online case), our proposed algorithm can achieve optimal complexity. This highlights the advantages of our algorithm in terms of parameter selection.

## 5.2. Convergence for R-Hybrid-SGD

**Lemma 5.7.** *Suppose that Assumptions 2.2 and 2.3 hold. Choose $v_0 = \text{grad} f_{\mathcal{B}_0}(x_0)$, $|\mathcal{B}_0| = b_0$, then the variance of the R-Hybrid-SGD estimator is bounded above by inequality (11), where $\mathcal{M}_1 = \tilde{L}^2$, $\mathcal{M}_2 = 1$, $\gamma_{k,i} = (1-\beta)^{2(k-i)}$, and $\lambda_k = (1-\beta)^{2(k-1)} \frac{\mathbb{1}_{\{b_0 < n\}}}{b_0} + \beta$.*

Building on Lemma 5.7 and Theorem 5.1, we obtain the following convergence result for R-Hybrid-SGD.

**Theorem 5.8.** *Suppose that Assumptions 2.2 and 2.3 hold. Then the output $x_\zeta$ after running $K$ iterations of Riemannian Hybrid-SGD algorithm satisfies $\mathbb{E}\left[\|\text{grad} f(x_\zeta)\|^2\right] \leqslant \frac{\varphi_3 \tilde{\Delta}_0}{\phi_3 K} + \frac{(\varphi_3 \eta + 4\phi_3)\sigma^2}{2\phi_3 K}\left(\frac{\mathbb{1}_{\{b_0<n\}}}{b_0 \beta} + K\beta\right)$, where $\varphi_3 = 2\left(\frac{\tilde{L}^2\eta^2}{\beta} + 1\right)$ and $\phi_3 = \eta\left(\frac{1-\eta L}{2} - \frac{\eta^2 \tilde{L}^2}{2\beta}\right)$.*

**Corollary 5.9.** *Suppose that Assumptions 2.2 and 2.3 hold. Set $\beta = K^{-2/3}$, $\eta = \min\left\{\frac{1}{2L}, \frac{\sqrt{\beta}}{2L}\right\}$. If we choose $b_0 = \min\{n, K^{1/3}\}$ under finite-sum case and $b_0 = K^{1/3}$ in online setting. Then the IFO complexity of Riemannian Hybrid-SGD to obtain $\varepsilon$-accurate solution is*

$$\begin{cases} \mathcal{O}\left(\min\{n, \varepsilon^{-1}\} + \varepsilon^{-3}\right), & \text{(finite-sum)} \\ \mathcal{O}\left(\varepsilon^{-3}\right), & \text{(online)} \end{cases}$$

## 5.3. Convergence for Other Riemannian Algorithms

### Convergence Analysis for R-SRM

**Theorem 5.10.** *Suppose that Assumptions 2.2 and 2.3 hold. Then the output $x_\zeta$ after running $K$ iterations of R-SRM algorithm satisfies $\mathbb{E}\left[\|\text{grad} f(x_\zeta)\|^2\right] \leqslant \frac{\varphi_4 \tilde{\Delta}_0}{\phi_4 K} + \frac{(\varphi_4 \eta + 4\phi_4)\sigma^2}{2\phi_4 K}\left(\frac{1}{b_0 \beta} + 2K\beta\right)$, where $\varphi_4 = 2\left(\frac{2\tilde{L}^2\eta^2}{\beta} + 1\right)$ and $\phi_4 = \eta\left(\frac{1-\eta L}{2} - \frac{\eta^2 \tilde{L}^2}{\beta}\right)$.*

**Corollary 5.11.** *Suppose that Assumptions 2.2 and 2.3 hold. Set $\beta = K^{-2/3}$, $\eta = \min\left\{\frac{1}{2L}, \frac{\sqrt{\beta}}{2\sqrt{2}\tilde{L}}\right\}$. If we choose $b_0 = \min\{n, K^{1/3}\}$ under finite-sum case and $b_0 = K^{1/3}$ in online setting. Then the IFO complexity of Riemannian Hybrid-SGD to obtain $\varepsilon$-accurate solution is*

$$\begin{cases} \mathcal{O}\left(\min\{n, \varepsilon^{-1}\} + \varepsilon^{-3}\right), & \text{(finite-sum)} \\ \mathcal{O}\left(\varepsilon^{-3}\right), & \text{(online)} \end{cases}$$

*Remark 5.12.* Although our proof requires that $v_0 = \text{grad} f_{\mathcal{B}_0}(x_0)$ be a batch gradient with a sufficiently large batch size, the complexity can be enhanced from $\tilde{\mathcal{O}}\left(\varepsilon^{-3}\right)$ (Han & Gao, 2021a) to $\mathcal{O}\left(\varepsilon^{-3}\right)$ in the online setting.

### Convergence Analysis for R-SVRG

**Theorem 5.13.** *Suppose that Assumptions 2.2 and 2.3 hold. Then the output $x_\zeta$ after running $K = Sm$ iterations of R-SVRG algorithm satisfies $\mathbb{E}\left[\|\text{grad} f(x_\zeta)\|^2\right] \leqslant$*

$\frac{\varphi_5 \tilde{\Delta}_0}{\phi_5 K} + \frac{(\varphi_5 \eta + 4\phi_5)\mathbb{1}_{\{b<n\}}\sigma^2}{2\phi_5 b}$, *where* $\varphi_5 = 2\left(\frac{\eta^2 m^2 \mathcal{L}}{b'} + 1\right)$, $\phi_5 = \eta\left(\frac{1-\eta L}{2} - \frac{\eta^2 m^2 \mathcal{L}}{2b'}\right)$.

**Corollary 5.14.** *Suppose that Assumptions 2.2 and 2.3 hold. Set $m = \sqrt{b'}$ and $\eta = \min\left\{\frac{1}{2L}, \frac{1}{2\tilde{L}\mu\nu}\right\}$. Choose $b = n$, $b' = n^{2/3}$ in finite-sum setting and $b = \Theta\left(\varepsilon^{-2}\right)$, $b' = \Theta\left(\varepsilon^{-4/3}\right)$ under online case, then the IFO complexity of R-SVRG to obtain $\varepsilon$-accurate solution is*

$$\begin{cases} \mathcal{O}\left(n^{2/3}\varepsilon^{-2}\right), & \text{(finite-sum)} \\ \mathcal{O}\left(\varepsilon^{-10/3}\right), & \text{(online)} \end{cases}$$

*Remark 5.15.* The IFO complexity result of the R-SVRG that we present in Theorem 5.13 matches the optimal complexity of the existing SVRG-type algorithms (Han & Gao, 2021b; Li et al., 2021).

### Convergence Analysis for R-PAGE

**Theorem 5.16.** *Suppose that Assumptions 2.2 and 2.3 hold. Then the output $x_\zeta$ after running $K$ iterations of R-PAGE algorithm satisfies $\mathbb{E}\left[\|\text{grad} f(x_\zeta)\|^2\right] \leqslant \frac{\varphi_6 \tilde{\Delta}_0}{\phi_6 K} + \frac{\varphi_6 \eta + 4\phi_6}{2\phi_6} \cdot \frac{\mathbb{1}_{\{b<n\}}\sigma^2}{b}$, where $\varphi_6 = 2\left(\frac{\tilde{L}^2\eta^2}{b'p} + 1\right)$ and $\phi_6 = \eta\left(\frac{1-\eta L}{2} - \frac{\tilde{L}^2\eta^2}{2b'p}\right)$.*

**Corollary 5.17.** *Suppose that Assumptions 2.2 and 2.3 hold. Set $p = \frac{b'}{b+b'}$, $b' \leqslant \sqrt{b}$ and $\eta = \min\left\{\frac{1}{2L}, \frac{b'}{2\tilde{L}\sqrt{b+b'}}\right\}$. Choose $b = n$ in the finite-sum setting and $b = \frac{18\sigma^2}{\varepsilon^2}$ under online case. Then the IFO complexity of Riemannian PAGE to obtain $\varepsilon$-accurate solution is*

$$\begin{cases} \mathcal{O}\left(n + \sqrt{n}\varepsilon^{-2}\right), & \text{(finite-sum)} \\ \mathcal{O}\left(\varepsilon^{-2} + \varepsilon^{-3}\right), & \text{(online)} \end{cases}$$

*Remark 5.18.* Although Demidovich et al. (2024) have proposed R-PAGE, they only provided convergence results under exponential map and parallel transport. To our knowledge, this is the first theoretical result on R-PAGE under retraction and vector transport in the non-convex setting. Furthermore, this result is consistent with the lower bounds for both finite-sum and online problems.

# 6. Experiments

In this section, we compare the proposed R-SRVRG, R-SVRRM, and R-Hybrid-SGD algorithms (dashed line) with several Riemannian algorithms (solid line) across different tasks, including R-SD, R-CG (Absil et al., 2008), R-SGD, R-SRM, R-SPIDER, R-AbaSRG, R-AbaSVRG, and R-PAGE. For our proposed algorithms and VR-based methods, a fixed step size is considered. We established an inner loop size $m = \sqrt{n}$ for both our methods and the VR methods, with

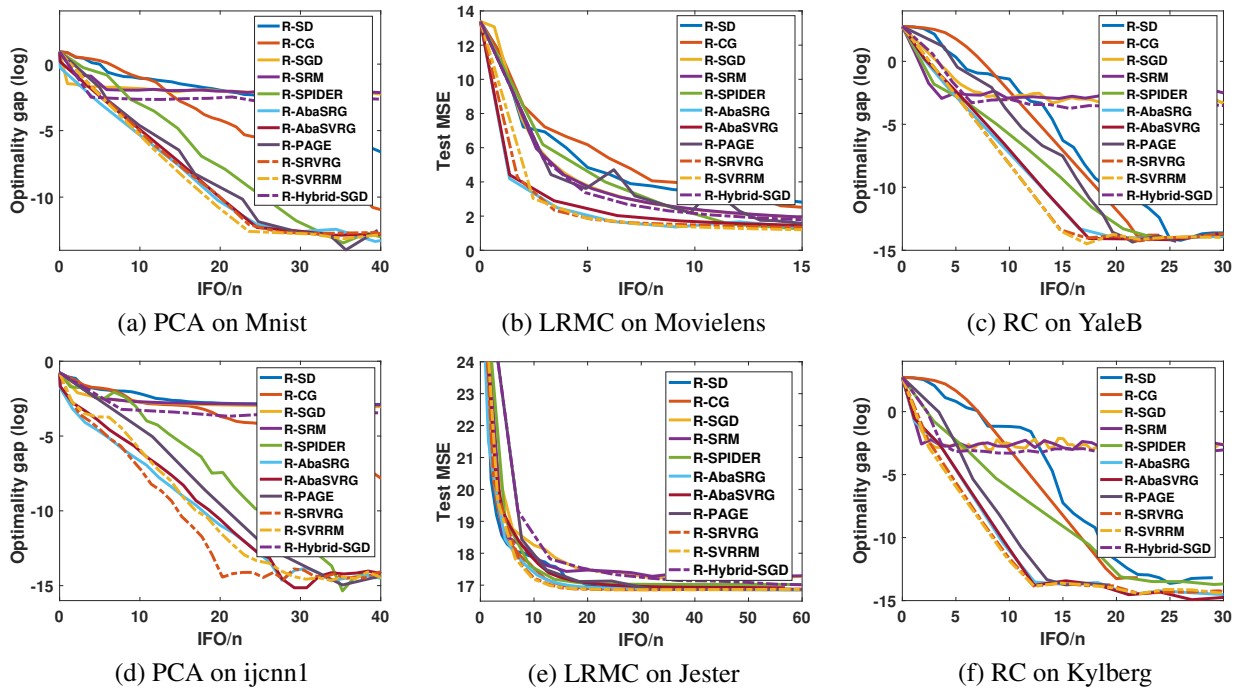

*Figure 1.* The comparison of the performance of our proposed algorithm (dashed line) with other methods (solid line) across different tasks and datasets.

a mini-batch size $b' = \sqrt{n}$ for the VR methods and $b' = 0.5\sqrt{n}$ for our methods. We selected the same parameters for R-SRVRG and R-SVRRM. Similarly, we chose identical parameters for R-SRM and R-Hybrid-SGD for comparison. Except for R-PAGE, the parameters of the other algorithms are referenced from (Han & Gao, 2021b). In Figure 1, the $x$-axis represents IFO (n), while the $y$-axis represents the optimality gap (which is precalculated) or test mean square error (MSE). All experiments are implemented in Matlab based on the code from the Man-Opt package (Boumal et al., 2014) and (Han & Gao, 2021b) [1] on a i7-1075H 2.6GHz CPU processor. The Appendix C presents more experimental results and details.

We consider PCA and LRMC problems on Grassmann manifold, and Riemannian centroid (RC) computation on symmetric positive definite (SPD) manifold. The experimental results demonstrate that our proposed R-SRVRG and R-SVRRM algorithms perform superiorly across multiple tasks. As an improvement over R-Hybrid-SGD, our R-SRVRG shows significantly better performance. Additionally, the selection of the parameter $\beta$ and the constant step size $\eta$ for the proposed R-SRVRG, R-SVRRM, and R-Hybrid-SGD algorithms demonstrate robustness across various scenarios (see Appendix C).

**PCA on Grassmann Manifold** The PCA problem in-

[1] https://github.com/andyjm3/R-AbaVR

volves minimizing the reconstruction error between the projected data points and the original data over the set of orthogonal projection matrices $\mathbf{U} \in \mathrm{St}(r, d)$, i.e., $\min_{\mathbf{U} \in \mathrm{St}(r,d)} \frac{1}{n} \sum_{i=1}^{n} \|x_i - \mathbf{U}\mathbf{U}^\top x_i\|_2^2$. In fact, PCA is equivalent to the following problem on the Grassmann manifold, $\min_{\mathbf{U} \in \mathrm{Gr}(r,d)} -\frac{1}{n} \sum_{i=1}^{n} \mathbf{x}_i^\top \mathbf{U}\mathbf{U}^\top \mathbf{x}_i$. Figures 1 (a) and (c) illustrate the performance of various algorithms in addressing the PCA problem on the MNIST dataset (LeCun et al., 1998) with $(n, d, r) = (60000, 784, 5)$ and ijcnn1 dataset from LibSVM (Chang & Lin, 2011) with $(n, d, r) = (49990, 22, 5)$, respectively. It can be observed that Our R-SRVRG and R-SVRRM algorithms outperform alternative methods, while the R-Hybrid-SGD algorithm demonstrates superior performance compared to the R-SRM algorithm.

**LRMC on Grassmann Manifold** Given a matrix $\mathrm{A} \in \mathbb{R}^{d \times n}$ with missing entries, the rank-$r$ matrix completion problem is to $\min_{\mathbf{U},\mathbf{V}} \|\mathcal{P}_\Omega(\mathbf{A}) - \mathcal{P}_\Omega(\mathbf{U}\mathbf{V})\|^2$, with $\mathbf{U} \in \mathbb{R}^{d \times r}, \mathbf{V} \in \mathbb{R}^{r \times n}$, where $\Omega$ is the set of indices for which we know the entries in $\mathbf{X}$, and the operator $\mathcal{P}_\Omega$ acts as $\mathcal{P}_\Omega(\mathbf{X}_{ij}) = \mathbf{X}_{ij}$ if $(i, j) \in \Omega$ and $\mathcal{P}_\Omega(\mathbf{X}_{ij}) = 0$ otherwise. Since the factorization into $\mathbf{U}$ and $\mathbf{V}$ is not unique and depends only on the column space of $\mathbf{U}$, the problem is defined on the Grassmann manifold $\mathrm{Gr}(r, d)$. Partitioning $\mathbf{a}_1, \ldots, \mathbf{a}_n$, it is equivalent to $\min_{\mathbf{U} \in \mathrm{Gr}(r,d), \mathbf{v}_i \in \mathbb{R}^r} \frac{1}{n} \sum_{i=1}^{n} \|\mathcal{P}_{\Omega_i}(\mathbf{a}_i) - \mathcal{P}_{\Omega_i}(\mathbf{U}\mathbf{v}_i)\|^2$, where $\mathcal{P}_{\Omega_i}$ is the sampling operator for the $i$-th column.

Given $\mathbf{U}, \mathbf{v}_i$ in this admits a closed form solution. Figures 1 (b) and (e) show that our proposed algorithms R-SRVRG and R-SVRRM are slightly better than that of all other algorithms on datasets Movielens-1M (Harper & Konstan, 2015) and Jester (Goldberg et al., 2001).

**RC on SPD Manifold** Given $n$ points $\{\mathbf{X}_1, \ldots, \mathbf{X}_n\} \in \mathcal{S}^d_{++}$. Let $\|\cdot\|_F$ denote the Frobenius norm, and $\log(\cdot)$ represent the principal matrix logarithm. Then the Riemannian centroid is derived from the solution to the problem $\min_{\mathbf{C} \in \mathcal{S}^d_{++}} \frac{1}{n} \sum_{i=1}^n \left\| \log\left(\mathbf{C}^{-1/2} \mathbf{X}_i \mathbf{C}^{-1/2}\right) \right\|_F^2$. We compare algorithms on Extended Yale B dataset (Wright et al., 2008) and Kylberg dataset (Kylberg, 2011). From Figure 1 (c) and (f), we observe that R-SRVRG and R-SVRRM still perform better compared to the other algorithms.

## 7. Conclusion

This paper first combines SVRG-type estimator with SARAH-type one to propose the SRVRG estimator and extend it to Riemannian manifolds. Subsequently, we introduce a unified variance reduced estimator, SRGE, and based on this, propose a unified framework for Riemannian stochastic variance reduction algorithms, named SERENA. Additionally, we provide a unified theoretical analysis for Riemannian stochastic variance reduction algorithms under retraction and vector transport, which is more general and efficient than using the exponential map and parallel transport. Theoretical results indicate that the IFO complexity of our proposed R-SRVRG, R-SVRRM, and R-Hybrid-SGD match the lower bound of complexity for non-convex stochastic optimization. Experimental results also demonstrate the superiority of our proposed algorithms. However, our current framework is limited to smooth problems. Therefore, future work will involve extending this framework and theoretical analysis to address non-smooth optimization challenges on Riemannian manifolds. Furthermore, Riemannian second-order methods have garnered increasing attention, such as the RNGD (Hu et al., 2024), R-SQN-VR (Kasai et al., 2018a), and R-SVRC (Zhang & Davanloo Tajbakhsh, 2023) algorithms. We will explore the integration of this framework with second-order information to enhance the convergence rate.

## Acknowledgments

This paper was supported in part by the National Key R&D Program of China (2022YFA1008300), the National Natural Science Foundation of China (12471308, 12101334, 62472415), the Guangdong Provincial Science and Technology Plan (2022B1515130009, 2025A1515010103), the Excellent Young Scholars of Shenzhen (RCYX20231211090247060), and the General Research Fund of Research Grants Council (15304721).

## Impact Statement

This paper presents work whose goal is to advance the field of Machine Learning. There are many potential societal consequences of our work, none which we feel must be specifically highlighted here.

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

# A. Specific Algorithms

Here, we list the algorithms proposed in this paper, including R-SRVRG, R-SVRRM, and R-Hybrid-SGD.

---
**Algorithm 2** Riemannian SRVRG
---
**Input:** Step size $\eta$, outer loop size $S$, inner loop size $m$, batch size $b$, $b'$, initial point $\tilde{x}_0$.

1: **for** $s = 1, 2, \ldots, S$ **do**
2: $\quad x_0^s = \tilde{x}_{s-1}$.
3: $\quad$ Sample $\mathcal{B}$ uniformly at random from $[n]$ of size $b$ and calculate $v_0^s = \operatorname{grad} f_{\mathcal{B}}(x_0^s)$.
4: $\quad x_1^s = R_{x_0^s}(-\eta v_0^s)$.
5: $\quad$ **for** $k = 1, \ldots, m - 1$ **do**
6: $\quad\quad$ Sample $\mathcal{I}_k$ and $\mathcal{J}_k$ independently from $[n]$.
7: $\quad\quad$ Calculate $v_k^s = (1 - \beta) \operatorname{grad} f_{\mathcal{I}_k}(x_k^s) + \beta \left( \operatorname{grad} f_{\mathcal{J}_k}(x_k^s) - \mathcal{T}_{x_0^s}^{x_k^s} \left( \operatorname{grad} f_{\mathcal{J}_k}(x_0^s) - v_0^s \right) \right) + (1 - \beta) \mathcal{T}_{x_{k-1}^s}^{x_k^s} \left( v_{k-1}^s - \operatorname{grad} f_{\mathcal{I}_k}(x_{k-1}^s) \right)$
8: $\quad\quad x_{k+1}^s = R_{x_k^s}(-\eta v_k^s)$.
9: $\quad$ **end for**
10: $\quad \tilde{x}_s = x_m^s$
11: **end for**
12: **Output:** $x_\zeta$ uniformly selected at random from $\{x_k^s\}$, $k \in [m - 1], s \in [S]$.

---
**Algorithm 3** Riemannian SVRRM
---
**Input:** Step size $\eta$, outer loop size $S$, inner loop size $m$, batch size $b$, $b'$, initial point $\tilde{x}_0$.

1: **for** $s = 1, 2, \ldots, S$ **do**
2: $\quad x_0^s = \tilde{x}_{s-1}$.
3: $\quad$ Sample $\mathcal{B}$ uniformly at random from $[n]$ of size $b$ and calculate $v_0^s = \operatorname{grad} f_{\mathcal{B}}(x_0^s)$.
4: $\quad x_1^s = R_{x_0^s}(-\eta v_0^s)$.
5: $\quad$ **for** $k = 1, \ldots, m - 1$ **do**
6: $\quad\quad$ Sample $\mathcal{I}_k$ randomly from $[n]$.
7: $\quad\quad$ Calculate $v_k^s = (1 - \beta) \operatorname{grad} f_{\mathcal{I}_k}(x_k^s) + \beta \left( \operatorname{grad} f_{\mathcal{I}_k}(x_k^s) - \mathcal{T}_{x_0^s}^{x_k^s} \left( \operatorname{grad} f_{\mathcal{I}_k}(x_0^s) - v_0^s \right) \right) + (1 - \beta) \mathcal{T}_{x_{k-1}^s}^{x_k^s} \left( v_{k-1}^s - \operatorname{grad} f_{\mathcal{I}_k}(x_{k-1}^s) \right)$
8: $\quad\quad x_{k+1}^s = R_{x_k^s}(-\eta v_k^s)$.
9: $\quad$ **end for**
10: $\quad \tilde{x}_s = x_m^s$
11: **end for**
12: **Output:** $x_\zeta$ uniformly selected at random from $\{x_k^s\}$, $k \in [m - 1], s \in [S]$.

---
**Algorithm 4** Riemannian Hybrid-SGD
---
**Input:** Step size $\eta$, mini-batch size $b_0$, the initial point $\tilde{x}_0$, $\beta$.

1: Sample $\mathcal{B}$ uniformly at random from $[n]$ of size $b_0$.
2: $v_0 = \operatorname{grad} f_{\mathcal{B}}(x_0)$.
3: $x_1 = R_{x_0}(-\eta v_0)$.
4: **for** $k = 1, \ldots, K - 1$ **do**
5: $\quad$ Sample $i_k$ and $j_k$ independently from $[n]$.
6: $\quad$ Calculate $v_k = \beta \operatorname{grad} f_{i_k}(x_k) + (1 - \beta) \operatorname{grad} f_{j_k}(x_k) + (1 - \beta) \mathcal{T}_{x_{k-1}}^{x_k} \left( v_{k-1} - \operatorname{grad} f_{j_k}(x_{k-1}) \right)$
7: $\quad x_{k+1} = R_{x_k}(-\eta v_k)$
8: **end for**
9: **Output:** $x_\zeta$ uniformly selected at random from $\{x_k\}_{k=0}^K$.

---

# B. Missing Proofs

## B.1. Useful Lemma

**Lemma B.1.** *(Retraction Lipschitzness with vector transport (Lemma A.2 in (Han & Gao, 2021b))). Suppose that the norm of gradient is bounded by $G$ (Assumption 2.2 (2)), $f$ is average retraction $L_l$-Lipschitz (Assumption 2.2 (5)), and the difference between parallel transport $P_y^x$ and vector transport $\mathcal{T}_y^x$ under same retraction is bounded (Assumption 2.2 (6)). Then for all $x, y = R_x(\xi) \in \mathcal{X}$,*

$$\mathbb{E}\left[\left\|\operatorname{grad} f_i(x) - \mathcal{T}_y^x \operatorname{grad} f_i(y)\right\|\right] \leqslant \tilde{L}\|\xi\|,$$

*where $\tilde{L} = (L_l + \theta G)$.*

**Lemma B.2.** *(Lemma 1 in (Han & Gao, 2021b)) Suppose Assumptions 2.2 and 2.3 hold and consider the estimator $v_k^s = \operatorname{grad} f_{\mathcal{B}'}(x_k^s) - \mathcal{T}_{x_0^s}^{x_k^s}(\operatorname{grad} f_{\mathcal{B}'}(x_0^s) - v_0^s)$, where $|\mathcal{B}'| = b'$ and $v_0^s = \operatorname{grad} f_{\mathcal{B}}(x_0^s)$, $|\mathcal{B}| = b$. We have*

$$\mathbb{E}\left[\|v_k^s - \operatorname{grad} f(x_k^s)\|^2\right] \leqslant \frac{k}{b'}\tilde{L}^2\mu^2\nu^2\eta^2\sum_{i=0}^{k-1}\mathbb{E}\left[\|v_i^s\|^2\right] + \mathbb{1}_{\{b<n\}}\frac{\sigma^2}{b},$$

*where $\tilde{L} = L_l + \theta G$.*

## B.2. The proof in Section 5

### B.2.1. THE PROOF OF THEOREM 5.1

*Proof.* In the $k$-th inner loop of $s$-th outer loop, by retraction $L$-smooth in Assumption 2.2 (4) , we have

$$
\begin{aligned}
f\left(x_{k+1}^s\right) &\leqslant f\left(x_k^s\right) - \left\langle\operatorname{grad} f\left(x_k^s\right), \eta v_k^s\right\rangle + \frac{\eta^2 L}{2}\|v_k^s\|^2 \\
&= f\left(x_k^s\right) - \eta\|v_k^s\|^2 + \eta\left\langle v_k^s - \operatorname{grad} f\left(x_k^s\right), v_k^s\right\rangle + \frac{\eta^2 L}{2}\|v_k^s\|^2 \\
&\leqslant f\left(x_k^s\right) - \eta\left(1 - \frac{\eta L}{2}\right)\|v_k^s\|^2 + \frac{\eta}{2}\|v_k^s\|^2 + \frac{\eta}{2}\|v_k^s - \operatorname{grad} f\left(x_k^s\right)\|^2 \\
&= f\left(x_k^s\right) - \eta\left(\frac{1 - \eta L}{2}\right)\|v_k^s\|^2 + \frac{\eta}{2}\|v_k^s - \operatorname{grad} f\left(x_k^s\right)\|^2,
\end{aligned}
$$

where the second inequality is due to the fact $2\langle a, b\rangle \leqslant \|a\|^2 + \|b\|^2$ for any vector $a, b \in \mathbb{R}^d$. Taking expectation of the above inequality and summing over $k = 0, \ldots, m-1$ obtain

$$
\begin{aligned}
\mathbb{E}\left[f\left(x_m^s\right)\right] &\leqslant \mathbb{E}\left[f\left(x_0^s\right)\right] - \eta\left(\frac{1 - \eta L}{2}\right)\sum_{k=0}^{m-1}\mathbb{E}\left[\|v_k^s\|^2\right] + \frac{\eta}{2}\sum_{k=0}^{m-1}\mathbb{E}\left[\|v_k^s - \operatorname{grad} f\left(x_k^s\right)\|^2\right] \\
&\leqslant \mathbb{E}\left[f\left(x_0^s\right)\right] - \eta\left(\frac{1 - \eta L}{2}\right)\sum_{k=0}^{m-1}\mathbb{E}\left[\|v_k^s\|^2\right] + \frac{\eta^3\mathcal{M}_1}{2}\sum_{k=0}^{m-1}\sum_{i=0}^{k-1}\gamma_{k,i}\mathbb{E}\left[\|v_i^s\|^2\right] + \frac{\eta\mathcal{M}_2}{2}\sum_{k=0}^{m-1}\lambda_k\sigma^2 \\
&\leqslant \mathbb{E}\left[f\left(x_0^s\right)\right] - \eta\left(\frac{1 - \eta L}{2}\right)\sum_{k=0}^{m-1}\mathbb{E}\left[\|v_k^s\|^2\right] + \frac{\eta^3\mathcal{M}_1}{2}\sum_{k=0}^{m-1}\Gamma_k^s\mathbb{E}\left[\|v_k^s\|^2\right] + \frac{\eta\mathcal{M}_2}{2}\sum_{k=0}^{m-1}\lambda_k\sigma^2,
\end{aligned}
$$

where the second inequality follows from (11) and the last inequality is due to the definition of $\Gamma_k^s = \sum_{k=0}^{m-1}\gamma_{k,0}$ and the fact $\sum_{k=0}^{m-1}\gamma_{k,0} \geqslant \sum_{i=0}^{k-1}\gamma_{k,i}$ for any $i \in [k], k \in [m-1]$. Telescoping this inequality over $s$ from 1 to $S$ and noting that $x_m^s = \tilde{x}_s = x_0^{s+1}$, we have

$$\mathbb{E}\left[f\left(x_m^S\right)\right] \leqslant \mathbb{E}\left[f\left(\tilde{x}_0\right)\right] - \eta\left(\frac{1 - \eta L}{2}\right)\sum_{s=1}^{S}\sum_{k=0}^{m-1}\mathbb{E}\left[\|v_k^s\|^2\right] + \frac{\eta^3\mathcal{M}_1}{2}\sum_{s=1}^{S}\sum_{k=0}^{m-1}\Gamma_k^s\mathbb{E}\left[\|v_k^s\|^2\right] + \frac{\eta\mathcal{M}_2}{2}\sum_{s=1}^{S}\sum_{k=0}^{m-1}\lambda_k\sigma^2.$$

By definition of $\hat{\Gamma}$ and $\hat{\Lambda}$, the following inequality is satisfied:

$$\mathbb{E}\left[f\left(x_m^S\right)\right] \leqslant \mathbb{E}\left[f\left(\tilde{x}_0\right)\right] - \eta\left(\frac{1-\eta L-\eta^2 \mathcal{M}_1 \hat{\Gamma}}{2}\right)\sum_{s=1}^{S}\sum_{k=0}^{m-1}\mathbb{E}\left[\|v_k^s\|^2\right] + \frac{\eta \mathcal{M}_2 \hat{\Lambda}\sigma^2}{2}.$$

Denote $\phi = \eta\left(\frac{1-\eta L-\eta^2 \mathcal{M}_1 \hat{\Gamma}}{2}\right)$ and $\tilde{\Delta}_0 = \mathbb{E}\left[f\left(\tilde{x}_0\right)\right] - f(x_*)$. Therefore,

$$\sum_{s=1}^{S}\sum_{k=0}^{m-1}\mathbb{E}\left[\|v_k^s\|^2\right] \leqslant \frac{\tilde{\Delta}_0}{\phi} + \frac{\eta \mathcal{M}_2 \hat{\Lambda}\sigma^2}{2\phi}. \tag{12}$$

Considering that $x_\zeta$ is chosen from $\{x_k^s\}_{s\in[S],k\in[m-1]}$ uniformly at random, we obtain

$$\begin{aligned}
Sm\mathbb{E}\left[\|\operatorname{grad} f\left(x_\zeta\right)\|^2\right] &= \sum_{s=1}^{S}\sum_{k=0}^{m-1}\mathbb{E}\left[\|\operatorname{grad} f\left(x_k^s\right)\|^2\right]\\
&\leqslant 2\sum_{s=1}^{S}\sum_{k=0}^{m-1}\mathbb{E}\left[\|\operatorname{grad} f\left(x_k^s\right)-v_k^s\|^2\right] + 2\sum_{s=1}^{S}\sum_{k=0}^{m-1}\mathbb{E}\left[\|v_k^s\|^2\right]\\
&\leqslant 2\sum_{s=1}^{S}\sum_{k=0}^{m-1}\left(\mathcal{M}_1\eta^2\sum_{i=0}^{k}\gamma_{k,i}\mathbb{E}\left[\|v_i^s\|^2\right] + \mathcal{M}_2\lambda_k\sigma^2\right) + 2\sum_{s=1}^{S}\sum_{k=0}^{m-1}\mathbb{E}\left[\|v_k^s\|^2\right]\\
&\leqslant 2\mathcal{M}_1\eta^2\hat{\Gamma}\sum_{s=1}^{S}\sum_{k=0}^{m-1}\mathbb{E}\left[\|v_k^s\|^2\right] + 2\mathcal{M}_2\hat{\Lambda}\sigma^2 + 2\sum_{s=1}^{S}\sum_{k=0}^{m-1}\mathbb{E}\left[\|v_k^s\|^2\right]\\
&= 2\left(\mathcal{M}_1\eta^2\hat{\Gamma}+1\right)\sum_{s=1}^{S}\sum_{k=0}^{m-1}\mathbb{E}\left[\|v_k^s\|^2\right] + 2\mathcal{M}_2\hat{\Lambda}\sigma^2\\
&\leqslant \frac{\varphi\tilde{\Delta}_0}{\phi} + \frac{\varphi\eta\mathcal{M}_2\hat{\Lambda}\sigma^2}{2\phi} + 2\mathcal{M}_2\hat{\Lambda}\sigma^2,
\end{aligned}$$

where the second inequality utilizes the (11), by definition of $\hat{\Gamma}$ and $\hat{\Lambda}$, the third inequality holds. The last equality is due to $\varphi = 2\left(\mathcal{M}_1\eta^2\hat{\Gamma}+1\right)$, and the last inequality follows (12). Dividing both sides by $K = Sm$ gives

$$\mathbb{E}\left[\|\operatorname{grad} f\left(x_\zeta\right)\|^2\right] \leqslant \frac{\varphi\tilde{\Delta}_0}{\phi K} + \frac{\varphi\eta\mathcal{M}_2\hat{\Lambda}\sigma^2}{2\phi K} + \frac{2\mathcal{M}_2\hat{\Lambda}\sigma^2}{K}.$$

This completes the proof. $\qquad\qquad\qquad\qquad\qquad\qquad\qquad\qquad\qquad\qquad\qquad\qquad\qquad\qquad\square$

### B.2.2. THE PROOF OF RIEMANNIAN SRVRG AND RIEMANNIAN SVRRM METHODS

**The proof of Lemma 5.2**

*Proof.* Let us first denote $\delta_k^s = \operatorname{grad} f_{\mathcal{I}_k}(x_k^s) - \operatorname{grad} f(x_k^s)$, $\vartheta_k^s = u_k^s - \operatorname{grad} f\left(x_k^s\right)$. Note that $u_k^s = \operatorname{grad} f_{\mathcal{J}_k}(x_k^s) - \mathcal{T}_{x_0^s}^{x_k^s}\left(\operatorname{grad} f_{\mathcal{J}_k}(x_0^s) - v_0^s\right)$, $\mathcal{I}_k$ and $\mathcal{J}_k$ are independent, $|\mathcal{I}| = |\mathcal{J}| = b'$. Obviously, $\mathbb{E}_{\mathcal{I}_k}[\delta_k^s - \mathcal{T}_{x_{k-1}^s}^{x_k^s}\delta_{k-1}^s] = 0$ and $\mathbb{E}_{\mathcal{J}_k}[\vartheta_k^s] = 0$. For Riemannian SRVRG (4), we have

$$\begin{aligned}
&\|\Delta_k^s\|^2\\
&= \left\|(1-\beta)\operatorname{grad} f_{\mathcal{I}_k}(x_k^s) + \beta u_k^s + (1-\beta)\mathcal{T}_{x_{k-1}^s}^{x_k^s}\left(v_{k-1}^s - \operatorname{grad} f_{\mathcal{I}_k}(x_{k-1}^s)\right) - \operatorname{grad} f\left(x_k^s\right)\right\|^2\\
&= \left\|(1-\beta)\mathcal{T}_{x_{k-1}^s}^{x_k^s}\Delta_{k-1}^s - (1-\beta)\mathcal{T}_{x_{k-1}^s}^{x_k^s}\delta_{k-1}^s + (1-\beta)\delta_k^s + \beta\vartheta_k^s\right\|^2\\
&= (1-\beta)^2\left\|\mathcal{T}_{x_{k-1}^s}^{x_k^s}\Delta_{k-1}^s\right\|^2 + (1-\beta)^2\left\|\delta_k^s - \mathcal{T}_{x_{k-1}^s}^{x_k^s}\delta_{k-1}^s\right\|^2 + \beta^2\|\vartheta_k^s\|^2\\
&\quad+ 2(1-\beta)^2\left\langle\mathcal{T}_{x_{k-1}^s}^{x_k^s}\Delta_{k-1}^s, \delta_k^s - \mathcal{T}_{x_{k-1}^s}^{x_k^s}\delta_{k-1}^s\right\rangle + 2(1-\beta)\beta\left\langle\mathcal{T}_{x_{k-1}^s}^{x_k^s}\Delta_{k-1}^s, \vartheta_k^s\right\rangle + 2(1-\beta)\beta\left\langle\delta_k^s - \mathcal{T}_{x_{k-1}^s}^{x_k^s}\delta_{k-1}^s, \vartheta_k^s\right\rangle.
\end{aligned}$$

Taking the expectation w.r.t. $\mathcal{I}_k$ conditioned on $\mathcal{J}_k$, we obtain

$$
\mathbb{E}_{i_k}\left[\|\Delta_k^s\|^2\right] = (1-\beta)^2\left[\left\|\mathcal{T}_{x_{k-1}^s}^{x_k^s}\Delta_{k-1}^s\right\|^2\right] + (1-\beta)^2\mathbb{E}_{\mathcal{I}_k}\left[\left\|\delta_k^s - \mathcal{T}_{x_{k-1}^s}^{x_k^s}\delta_{k-1}^s\right\|^2\right] + \beta^2\left[\|\vartheta_k^s\|^2\right]
$$
$$
+ 2(1-\beta)\beta\left\langle\mathcal{T}_{x_{k-1}^s}^{x_k^s}\Delta_{k-1}^s, \vartheta_k^s\right\rangle.
$$

Taking the expectation w.r.t. $\mathcal{J}_k$ and noting that $\mathbb{E}_{(\mathcal{I}_k, \mathcal{J}_k)}[\cdot] = \mathbb{E}_{\mathcal{J}_k}\left[\mathbb{E}_{\mathcal{I}_k}\left[\cdot|\mathcal{J}_k\right]\right]$, $\mathbb{E}_{\mathcal{J}_k}[\vartheta_k^s] = 0$, we have

$$
\mathbb{E}_{(\mathcal{I}_k, \mathcal{J}_k)}\left[\|\Delta_k^s\|^2\right] = (1-\beta)^2\left[\left\|\mathcal{T}_{x_{k-1}^s}^{x_k^s}\Delta_{k-1}^s\right\|^2\right] + (1-\beta)^2\mathbb{E}_{\mathcal{I}_k}\left[\left\|\delta_k^s - \mathcal{T}_{x_{k-1}^s}^{x_k^s}\delta_{k-1}^s\right\|^2\right] + \beta^2\mathbb{E}_{\mathcal{J}_k}\left[\|\vartheta_k^s\|^2\right].
$$

Then taking the expectation over all the randomness and using the Lemma B.2, we obtain

$$
\mathbb{E}\left[\|\Delta_k^s\|^2\right] = (1-\beta)^2\mathbb{E}\left[\left\|\mathcal{T}_{x_{k-1}^s}^{x_k^s}\Delta_{k-1}^s\right\|^2\right] + (1-\beta)^2\mathbb{E}\left[\left\|\delta_k^s - \mathcal{T}_{x_{k-1}^s}^{x_k^s}\delta_{k-1}^s\right\|^2\right]
$$
$$
+ \beta^2\left(\frac{k\mathcal{L}\eta^2}{b'}\sum_{i=0}^{k-1}\mathbb{E}\left[\|v_i^s\|^2\right] + \mathbb{1}_{\{b<n\}}\frac{\sigma^2}{b}\right).
$$

Noting that

$$
\mathbb{E}\left[\left\|\delta_k^s - \mathcal{T}_{x_{k-1}^s}^{x_k^s}\delta_{k-1}^s\right\|^2\right] = \mathbb{E}\left[\left\|\operatorname{grad}f_{\mathcal{I}_k}(x_k^s) - \mathcal{T}_{x_{k-1}^s}^{x_k^s}\operatorname{grad}f_{\mathcal{I}_k}(x_{k-1}^s) - \left(\operatorname{grad}f(x_k^s) - \mathcal{T}_{x_{k-1}^s}^{x_k^s}\operatorname{grad}f(x_{k-1}^s)\right)\right\|^2\right].
$$

By using the fact $\mathbb{E}[\|x - \mathbb{E}[x]\|^2] \leqslant \mathbb{E}[\|x\|^2]$, we have

$$
\mathbb{E}\left[\|\Delta_k^s\|^2\right] \leqslant (1-\beta)^2\mathbb{E}\left[\|\Delta_{k-1}^s\|^2\right] + \frac{(1-\beta)^2}{b'}\mathbb{E}\left[\left\|\operatorname{grad}f_{i_k}(x_k^s) - \mathcal{T}_{x_{k-1}^s}^{x_k^s}\operatorname{grad}f_{i_k}(x_{k-1}^s)\right\|^2\right]
$$
$$
+ \beta^2\left(\frac{k\mathcal{L}\eta^2}{b'}\sum_{i=0}^{k-1}\mathbb{E}\left[\|v_i^s\|^2\right] + \mathbb{1}_{\{b<n\}}\frac{\sigma^2}{b}\right).
$$

By applying the Lemma B.1, we have

$$
\mathbb{E}\left[\|\Delta_k^s\|^2\right] \leqslant (1-\beta)^2\mathbb{E}\left[\|\Delta_{k-1}^s\|^2\right] + \frac{(1-\beta)^2}{b'}\tilde{L}^2\eta^2\mathbb{E}\left[\|v_{k-1}^s\|^2\right] + \beta^2\left(\frac{k\mathcal{L}\eta^2}{b'}\sum_{i=0}^{k-1}\mathbb{E}\left[\|v_i^s\|^2\right] + \mathbb{1}_{\{b<n\}}\frac{\sigma^2}{b}\right).
$$

Recursively applying this inequality gives

$$
\mathbb{E}\left[\|\Delta_k^s\|^2\right] \leqslant \frac{\tilde{L}^2\eta^2}{b'}\sum_{i=0}^{k-1}(1-\beta)^{2(k-i-1)}\mathbb{E}\left[\|v_i^s\|^2\right] + \frac{\beta^2 k\mathcal{L}\eta^2}{b'}\sum_{i=0}^{k-1}(1-\beta)^{2(k-i-1)}\mathbb{E}\left[\|v_i^s\|^2\right]
$$
$$
+ (1-\beta)^{2k}\mathbb{1}_{\{b<n\}}\frac{\sigma^2}{b} + \beta^2\sum_{i=0}^{k-1}(1-\beta)^{2(k-i-1)}\mathbb{1}_{\{b<n\}}\frac{\sigma^2}{b}
$$
$$
\leqslant \frac{\tilde{L}^2\eta^2}{b'}\sum_{i=0}^{k-1}(1-\beta)^{2(k-i-1)}\mathbb{E}\left[\|v_i^s\|^2\right] + \frac{\beta^2 m^2\mathcal{L}\eta^2}{b'}\sum_{i=0}^{k-1}(1-\beta)^{2(k-i-1)}\mathbb{E}\left[\|v_i^s\|^2\right]
$$
$$
+ \left((1-\beta)^{2k} + \beta\right)\mathbb{1}_{\{b<n\}}\frac{\sigma^2}{b}.
$$

For Riemannian SVRRM, by the difinition of $v_k^s$ in (5), we have

$$
\mathbb{E}\left[\|v_k^s - \operatorname{grad}f(x_k^s)\|^2\right]
$$
$$
= \mathbb{E}\left[\left\|(1-\beta)\operatorname{grad}f_{\mathcal{I}_k}(x_k^s) + \beta u_k^s + (1-\beta)\mathcal{T}_{x_{k-1}^s}^{x_k^s}\left(v_{k-1}^s - \operatorname{grad}f_{\mathcal{I}_k}(x_{k-1}^s)\right) - \operatorname{grad}f(x_k^s)\right\|^2\right]
$$

$$= (1-\beta)^2 \mathbb{E}\left[\left\|\mathcal{T}_{x_{k-1}^s}^{x_k^s}\left(v_{k-1}^s - \operatorname{grad} f(x_{k-1}^s)\right)\right\|^2\right]$$

$$+ \mathbb{E}\left[\left\|(1-\beta)\left(\operatorname{grad} f_{\mathcal{I}_k}(x_k^s) - \operatorname{grad} f(x_k^s) - \mathcal{T}_{x_{k-1}^s}^{x_k^s}\left(\operatorname{grad} f_{\mathcal{I}_k}(x_{k-1}^s) - \operatorname{grad} f(x_{k-1}^s)\right)\right) + \beta\left(u_k^s - \operatorname{grad} f(x_k^s)\right)\right\|^2\right]$$

$$\leqslant (1-\beta)^2 \mathbb{E}\left[\left\|v_{k-1}^s - \operatorname{grad} f\left(x_{k-1}^s\right)\right\|^2\right] + 2\beta^2 \mathbb{E}\left[\left\|u_k^s - \operatorname{grad} f(x_k^s)\right\|^2\right]$$

$$+ 2(1-\beta)^2 \mathbb{E}\left[\left\|\operatorname{grad} f_{\mathcal{I}_k}(x_k^s) - \operatorname{grad} f(x_k^s) - \mathcal{T}_{x_{k-1}^s}^{x_k^s}\left(\operatorname{grad} f_{\mathcal{I}_k}(x_{k-1}^s) - \operatorname{grad} f(x_{k-1}^s)\right)\right\|^2\right]$$

$$\leqslant (1-\beta)^2 \mathbb{E}\left[\left\|v_{k-1}^s - \operatorname{grad} f\left(x_{k-1}^s\right)\right\|^2\right] + 2(1-\beta)^2 \mathbb{E}\left[\left\|\operatorname{grad} f_{\mathcal{I}_k}(x_k^s) - \mathcal{T}_{x_{k-1}^s}^{x_k^s} \operatorname{grad} f_{\mathcal{I}_k}(x_{k-1}^s)\right\|^2\right]$$

$$+ 2\beta^2 \mathbb{E}\left[\left\|u_k^s - \operatorname{grad} f(x_k^s)\right\|^2\right]$$

$$\leqslant (1-\beta)^2 \mathbb{E}\left[\left\|v_{k-1}^s - \operatorname{grad} f\left(x_{k-1}^s\right)\right\|^2\right] + \frac{2(1-\beta)^2 \tilde{L}^2 \eta^2}{b'} \mathbb{E}\left[\left\|v_{k-1}^s\right\|^2\right] + 2\beta^2 \mathbb{E}\left[\left\|u_k^s - \operatorname{grad} f(x_k^s)\right\|^2\right]$$

$$\leqslant (1-\beta)^2 \mathbb{E}\left[\left\|v_{k-1}^s - \operatorname{grad} f\left(x_{k-1}^s\right)\right\|^2\right] + \frac{2(1-\beta)^2 \tilde{L}^2 \eta^2}{b'} \mathbb{E}\left[\left\|v_{k-1}^s\right\|^2\right] + \frac{2k\beta^2 \mathcal{L}\eta^2}{b'} \sum_{i=0}^{k-1} \mathbb{E}\left[\left\|v_i^s\right\|^2\right] + 2\beta^2 \mathbb{1}_{\{b<n\}} \frac{\sigma^2}{b},$$

where the second equality holds due to the unbiasedness of $\operatorname{grad} f_{i_k}(x_k)$, $\operatorname{grad} f_{i_k}(x_{k-1})$ and $\mathbb{E}[u_k^s] = \operatorname{grad} f(x_k^s)$. The first inequality is due to the fact $\|a+b\|^2 \leqslant 2\|a\|^2 + 2\|b\|^2$, the second inequality follows $\mathbb{E}[\|x - \mathbb{E}[x]\|^2] \leqslant \mathbb{E}[\|x\|^2]$. By using the Lemma B.1, we have the third inequality, the last is due to the Lemma B.2.

Recursively applying this inequality, we have

$$\mathbb{E}\left[\|\Delta_k^s\|^2\right] \leqslant \frac{2\tilde{L}^2 \eta^2}{b'} \sum_{i=0}^{k-1} (1-\beta)^{2(k-i-1)} \mathbb{E}\left[\|v_i^s\|^2\right] + \frac{2\beta^2 m^2 \mathcal{L}\eta^2}{b'} \sum_{i=0}^{k-1} (1-\beta)^{2(k-i-1)} \mathbb{E}\left[\|v_i^s\|^2\right]$$

$$+ \left((1-\beta)^{2k} + 2\beta\right) \mathbb{1}_{\{b<n\}} \frac{\sigma^2}{b}.$$

$\square$

### The proof of Theorem 5.3 and Corollary 5.4

*Proof.* Given the similarities in the theoretical analysis of R-SRVRG and R-SVRRM, we will use R-SRVRG as an example to present the proof. Note that $\mathcal{M}_1 = \frac{\tilde{L}^2 + \beta^2 \mathcal{L}m^2}{b'}$, $\mathcal{M}_2 = \frac{\mathbb{1}_{\{b<n\}}}{b}$, $\gamma_{k,i} = (1-\beta)^{2(k-i-1)}$, and $\lambda_k = (1-\beta)^{2k} + \beta$. Therefore, we have

$$\Gamma_k^s = \sum_{k=0}^{m-1} \gamma_{k,0} \leqslant \sum_{k=0}^{m-1} (1-\beta)^{2(k-1)} \leqslant \frac{1}{1-(1-\beta)^2} \leqslant \frac{1}{\beta} = \hat{\Gamma},$$

$$\sum_{s=1}^{S} \sum_{k=0}^{m-1} \lambda_k = \frac{1}{\beta} + K\beta = \hat{\Lambda}.$$

According to Theorem 5.1, we have

$$\mathbb{E}\left[\|\operatorname{grad} f(x_\zeta)\|^2\right] \leqslant \frac{\varphi_1 \tilde{\Delta}_0}{\phi_1 K} + \frac{(\varphi_1 \eta + 4\phi_1)\sigma^2}{2\phi_1 K b}\left(\frac{1}{\beta} + K\beta\right),$$

where $\varphi_1 = \frac{2\eta^2(\tilde{L}^2 + \beta^2 \mathcal{L}m^2)}{\beta b'} + 2$, $\phi_1 = \eta\left(\frac{1-\eta L}{2} - \frac{\eta^2(\tilde{L}^2 + \beta^2 \mathcal{L}m^2)}{2\beta b'}\right)$. Setting $\beta = m^{-1}, \eta = \min\{\frac{1}{2L}, \frac{\sqrt{b'}}{2\tilde{L}\sqrt{m}\sqrt{1+\mu^2\nu^2}}\}$, then we can derive that $2 \leqslant \varphi_1 \leqslant \frac{5}{2}$, $\frac{\eta}{8} \leqslant \phi_1 \leqslant \frac{\eta}{2}$.

In finite-sum case, if we choose $b = \min\left\{n, \sigma^2 \varepsilon^{-2}\right\}$ and $\sqrt{\frac{\beta}{b'}} = \max\left\{\frac{1}{\sqrt{n}}, \varepsilon\right\}$, we observe that $\sqrt{\frac{\beta}{b'}} \geqslant \varepsilon^2$. Consequently, we have $\mathbb{E}\left[\|\operatorname{grad} f(x_\zeta)\|^2\right] \leqslant \frac{20\tilde{\Delta}_0}{\eta K} + \frac{18\sigma^2}{Kb}\left(m + \frac{K}{m}\right) \leqslant \frac{20\tilde{\Delta}_0}{\eta K} + \frac{36\sigma^2}{b}$. To obtain $\varepsilon$-accurate solution, we require

$K = \Theta\left(\frac{1}{\sqrt{b'}\beta\varepsilon^2}\right)$, which is derived from the fact $\mathbb{E}\left[\|\operatorname{grad} f(x_\zeta)\|\right] \leqslant \sqrt{\mathbb{E}\left[\|\operatorname{grad} f(x_\zeta)\|^2\right]}$. Therefore, the total IFO complexity is given by $S(b + 3mb') = \frac{\sqrt{\beta}}{\sqrt{b'}\varepsilon^2}\left(\min\left\{n, \sigma^2\varepsilon^{-2}\right\} + \frac{3b'}{\beta}\right) = \mathcal{O}\left(\min\left\{n^{1/2}\varepsilon^{-2}, \varepsilon^{-3}\right\}\right)$.

Under online setting, denote $b = 72\sigma^2\varepsilon^{-2}$ and $\sqrt{\frac{\beta}{b'}} = \varepsilon$, then we have $\mathbb{E}\left[\|\operatorname{grad} f(x_\zeta)\|^2\right] \leqslant \frac{20\tilde{\Delta}_0}{\eta K} + \frac{36\sigma^2}{Kb} \leqslant \varepsilon^2$. To obtain $\varepsilon$-accurate solution, we require $K = \Theta\left(\frac{1}{\sqrt{b'}\beta\varepsilon^2}\right)$. Consequently, the total IFO complexity is given by $S(b + 3mb') = \frac{\sqrt{\beta}}{\sqrt{b'}\varepsilon^2}\left(b + \frac{3b'}{\beta}\right) = \mathcal{O}\left(\varepsilon^{-3}\right)$. The proof of Riemannian SVRRM is similar to R-SRVRG, thus we omitted it. □

### B.2.3. THE PROOF OF RIEMANNIAN HYBRID-SGD

**The proof of Lemma 5.7**

*Proof.* Let us first denote $\Delta_k = v_k - \operatorname{grad} f(x_k)$, $\delta_{k,i} = \operatorname{grad} f_{i_k}(x_k) - \operatorname{grad} f(x_k)$ and $\delta_{k,j} = \operatorname{grad} f_{j_k}(x_k) - \operatorname{grad} f(x_k)$.

$$\mathbb{E}\left[\|\Delta_k\|^2\right] = \mathbb{E}\left[\left\|\beta\operatorname{grad} f_{i_k}(x_k) + (1-\beta)\operatorname{grad} f_{j_k}(x_k) + (1-\beta)\mathcal{T}_{x_{k-1}}^{x_k}\left(v_{k-1} - \operatorname{grad} f_{j_k}(x_{k-1})\right) - \operatorname{grad} f(x_k)\right\|^2\right]$$

$$= \mathbb{E}\left[\left\|(1-\beta)\mathcal{T}_{x_{k-1}}^{x_k}\Delta_{k-1} + (1-\beta)\left(\delta_{k,j} - \mathcal{T}_{x_{k-1}}^{x_k}\delta_{k-1,j}\right) + \beta\delta_{k,i}\right\|^2\right]$$

$$= (1-\beta)^2\mathbb{E}\left[\|\Delta_{k-1}\|^2\right] + \mathbb{E}\left[\left\|(1-\beta)\left(\delta_{k,j} - \mathcal{T}_{x_{k-1}}^{x_k}\delta_{k-1,j}\right) + \beta\delta_{k,i}\right\|^2\right]$$

$$= (1-\beta)^2\mathbb{E}\left[\|\Delta_{k-1}\|^2\right] + (1-\beta)^2\mathbb{E}\left[\left\|\delta_{k,j} - \mathcal{T}_{x_{k-1}}^{x_k}\delta_{k-1,j}\right\|^2\right] + \beta^2\mathbb{E}\left[\|\delta_{k,i}\|^2\right]$$

$$\leqslant (1-\beta)^2\mathbb{E}\left[\|\Delta_{k-1}\|^2\right] + (1-\beta)^2\mathbb{E}\left[\left\|\mathcal{T}_{x_{k-1}}^{x_k}\operatorname{grad} f_{j_k}(x_{k-1}) - \operatorname{grad} f_{j_k}(x_k)\right\|^2\right] + \beta^2\mathbb{E}\left[\|\delta_{k,i}\|^2\right]$$

$$\leqslant (1-\beta)^2\mathbb{E}\left[\|\Delta_{k-1}\|^2\right] + (1-\beta)^2\eta^2\tilde{L}^2\mathbb{E}\left[\|v_{k-1}\|^2\right] + \beta^2\sigma^2,$$

where the first inequality is due to the fact $\mathbb{E}[\|x - \mathbb{E}[x]\|^2] \leqslant \mathbb{E}[\|x\|^2]$ and $\delta_{k,j} - \mathcal{T}_{x_{k-1}}^{x_k}\delta_{k-1,j} = \operatorname{grad} f_{j_k}(x_k) - \mathcal{T}_{x_{k-1}}^{x_k}\operatorname{grad} f_{j_k}(x_{k-1}) - \left(\operatorname{grad} f(x_k) - \mathcal{T}_{x_{k-1}}^{x_k}\operatorname{grad} f(x_{k-1})\right)$, while the last inequality is derived from Lemma B.2. Therefore, by applying this inequality recursively, we have

$$\mathbb{E}\left[\|\Delta_k\|^2\right] \leqslant (1-\beta)^{2(k-1)}\mathbb{E}\left[\|\Delta_0\|^2\right] + \eta^2\tilde{L}^2\sum_{i=0}^{k-1}(1-\beta)^{2(k-i)}\mathbb{E}\left[\|v_i\|^2\right] + \beta^2\sum_{i=0}^{k-1}(1-\beta)^{2(k-i-1)}\sigma^2.$$

Note that $v_0 = \operatorname{grad} f_{\mathcal{B}_0}(x_0)$ and $|\mathcal{B}_0| = b_0$, then we obtain

$$\mathbb{E}\left[\|\Delta_k\|^2\right] \leqslant \eta^2\tilde{L}^2\sum_{i=0}^{k-1}(1-\beta)^{2(k-i)}\mathbb{E}\left[\|v_i\|^2\right] + \beta^2\sum_{i=0}^{k-1}(1-\beta)^{2(k-i-1)}\sigma^2 + (1-\beta)^{2(k-1)}\frac{\mathbb{1}_{\{b_0<n\}}\sigma^2}{b_0}$$

$$\leqslant \eta^2\tilde{L}^2\sum_{i=0}^{k-1}(1-\beta)^{2(k-i)}\mathbb{E}\left[\|v_i\|^2\right] + \beta\sigma^2 + (1-\beta)^{2(k-1)}\frac{\mathbb{1}_{\{b_0<n\}}\sigma^2}{b_0},$$

where the first inequality is due to $\mathbb{E}[\|v_0 - \operatorname{grad} f(x_0)\|^2] = \mathbb{E}[\|\operatorname{grad} f_{\mathcal{B}_0}(x_0) - \operatorname{grad} f(x_0)\|^2] = \mathbb{1}_{\{b_0<n\}}\frac{\sigma^2}{b_0}$, the last inequality follows that $\beta^2\sum_{i=0}^{k-1}(1-\beta)^{2(k-i-1)} \leqslant \frac{\beta^2}{1-(1-\beta)^2} \leqslant \frac{\beta}{2-\beta} \leqslant \beta$, the proof is completed. □

**The proof of Theorem 5.8 and Corollary 5.9**

*Proof.* By Lemma 5.7, we know that $\mathcal{M}_1 = \tilde{L}^2$, $\mathcal{M}_2 = 1$, $\gamma_{k,i} = (1-\beta)^{2(k-i)}$, and $\lambda_k = (1-\beta)^{2(k-1)}\frac{\mathbb{1}_{\{b_0<n\}}}{b_0} + \beta$. Therefore,

$$\Gamma_k^s = \sum_{k=0}^{m-1}\gamma_{k,0} \leqslant \frac{1}{1-(1-\beta)^2} \leqslant \frac{1}{\beta} = \hat{\Gamma},$$

$$\sum_{k=0}^{K-1} \lambda_k \leqslant \frac{\mathbb{1}_{\{b_0<n\}}}{b_0 \beta (2-\beta)} + K\beta \leqslant \frac{\mathbb{1}_{\{b_0<n\}}}{b_0 \beta} + K\beta = \hat{\Lambda}.$$

According to Theorem 5.1, we have

$$\mathbb{E}\Big[\|\mathrm{grad}\, f(x_\zeta)\|^2\Big] \leqslant \frac{\varphi_3 \tilde{\Delta}_0}{\phi_3 K} + \frac{(\varphi_3 \eta + 4\phi_3)\sigma^2}{2\phi_3 K}\left(\frac{\mathbb{1}_{\{b_0<n\}}}{b_0 \beta} + K\beta\right),$$

where $\phi_3 = \eta\left(\frac{1-\eta L}{2} - \frac{\eta^2 \tilde{L}^2}{2\beta}\right)$ and $\varphi_3 = 2\left(\frac{2\tilde{L}^2 \eta^2}{\beta} + 1\right)$. Set $\beta = K^{-2/3}$ and $\eta = \min\left\{\frac{1}{2L}, \frac{\sqrt{\beta}}{2\tilde{L}}\right\}$, then $2 \leqslant \varphi_3 \leqslant 3$, $\frac{\eta}{8} \leqslant \phi_3 \leqslant \frac{\eta}{2}$. Therefore,

$$\mathbb{E}\Big[\|\mathrm{grad}\, f(x_\zeta)\|^2\Big] \leqslant \frac{24\tilde{\Delta}_0}{\eta K} + \frac{20\sigma^2}{K^{2/3}} + \frac{20\mathbb{1}_{\{b_0<n\}}\sigma^2}{K^{1/3}b_0}.$$

Note that $\eta = \Theta\left(K^{-1/3}\right)$, if we set $b_0 = \min\{n, K^{1/3}\}$ under finite-sum case, then we have $\mathbb{E}\Big[\|\mathrm{grad}\, f(x_\zeta)\|^2\Big] \leqslant \frac{24\tilde{\Delta}_0}{\eta K} + \frac{40\sigma^2}{K^{2/3}}$. To obtain $\varepsilon$-accurate solution, we require $K = \Theta\left(\varepsilon^{-3}\right)$, thus the total IFO complexity is given by $b_0 + 3K = \mathcal{O}\left(\min\{n, \varepsilon^{-1}\} + \varepsilon^{-3}\right)$. In online setting, set $b_0 = K^{1/3}$, then the total IFO complexity is given by $b_0 + 3K = \mathcal{O}\left(\varepsilon^{-3}\right)$. $\quad\square$

### B.2.4. THE PROOF OF OTHER RIEMANNIAN STOCHASTIC ALGORITHMS

**The proof for R-SRM Algorithm**. The upper bound of variance for R-SRM is as Lemma B.3. The Lemma B.3 states that $\mathcal{M}_1 = 2\tilde{L}^2$, $\mathcal{M}_2 = 2$, $\gamma_{k,i} = (1-\beta)^{2(k-i)}$, and $\lambda_k = (1-\beta)^{2(k-1)}\frac{\mathbb{1}_{\{b_0<n\}}}{b_0} + 2\beta$. Based on this, we obtain the convergence result 5.10 for R-SRM (The proof is similar to Theorem 5.8, thus we omitted it).

**Lemma B.3.** *Suppose that Assumptions 2.2 and 2.3 hold. Choose $v_0 = \mathrm{grad}\, f_{\mathcal{B}_0}(x_0)$, $|\mathcal{B}_0| = b_0$, then the estimator of R-SRM (8) satisfies*

$$\mathbb{E}\Big[\|\Delta_k\|^2\Big] \leqslant 2\eta^2 \tilde{L}^2 \sum_{i=0}^{k-1} (1-\beta)^{2(k-i)} \mathbb{E}\Big[\|v_i\|^2\Big] + 2\beta\sigma^2 + (1-\beta)^{2(k-1)}\frac{\mathbb{1}_{\{b_0<n\}}\sigma^2}{b_0}.$$

*Proof.* Define $\delta_k = \mathrm{grad}\, f_{i_k}(x_k) - \mathrm{grad}\, f(x_k)$.

$$\mathbb{E}\Big[\|v_k - \nabla f(x_k)\|^2\Big]$$

$$= \mathbb{E}\left[\left\|\nabla f_{i_k}(x_k) + (1-\beta)\mathcal{T}_{x_{k-1}}^{x_k}(v_{k-1} - \nabla f_{i_k}(x_{k-1})) - \nabla f(x_k)\right\|^2\right]$$

$$= \mathbb{E}\left[\left\|(1-\beta)\mathcal{T}_{x_{k-1}}^{x_k}(v_{k-1} - \nabla f(x_{k-1})) - (1-\beta)\left(\mathcal{T}_{x_{k-1}}^{x_k}\delta_{k-1} - \delta_k\right) + \beta\delta_k\right\|^2\right]$$

$$= (1-\beta)^2 \mathbb{E}\Big[\|v_{k-1} - \nabla f(x_{k-1})\|^2\Big] + \mathbb{E}\left[\left\|(1-\beta)\left(\mathcal{T}_{x_{k-1}}^{x_k}\delta_{k-1} - \delta_k\right) + \beta\delta_k\right\|^2\right]$$

$$\leqslant (1-\beta)^2 \mathbb{E}\Big[\|v_{k-1} - \nabla f(x_{k-1})\|^2\Big] + 2(1-\beta)^2 \mathbb{E}\left[\left\|\mathcal{T}_{x_{k-1}}^{x_k}\delta_{k-1} - \delta_k\right\|^2\right] + 2\beta^2 \mathbb{E}\Big[\|\delta_k\|^2\Big]$$

$$\leqslant (1-\beta)^2 \mathbb{E}\Big[\|v_{k-1} - \nabla f(x_{k-1})\|^2\Big] + 2(1-\beta)^2 \mathbb{E}\left[\left\|\mathcal{T}_{x_{k-1}}^{x_k}\nabla f_{i_k}(x_{k-1}) - \nabla f_{i_k}(x_k)\right\|^2\right] + 2\beta^2 \mathbb{E}\Big[\|\delta_k\|^2\Big]$$

$$\leqslant (1-\beta)^2 \mathbb{E}\Big[\|v_{k-1} - \nabla f(x_{k-1})\|^2\Big] + 2(1-\beta)^2 \eta^2 \tilde{L}^2 \mathbb{E}\Big[\|v_{k-1}\|^2\Big] + 2\beta^2 \sigma^2,$$

where the third equality is due to $\mathbb{E}\left[\mathcal{T}_{x_{k-1}}^{x_k}\delta_{k-1} - \delta_k\right] = 0$ and $\mathbb{E}[\delta_k] = 0$. The first inequality follows that $\|a + b\|^2 \leqslant 2\|a\|^2 + 2\|b\|^2$, the second follows from $\mathbb{E}\left[\mathcal{T}_{x_{k-1}}^{x_k}\nabla f_{i_k}(x_{k-1}) - \nabla f_{i_k}(x_k)\right] = \mathcal{T}_{x_{k-1}}^{x_k}\nabla f(x_{k-1}) - \nabla f(x_k)$ and $\mathbb{E}[\|x - \mathbb{E}[x]\|^2] \leqslant \mathbb{E}[\|x\|^2]$. By applying Lemma B.2, the last inequality holds true. Recursively applying this inequality and noting that $v_0 = \mathrm{grad}\, f_{\mathcal{B}_0}(x_0)$, $|\mathcal{B}_0| = b_0$, we can reach the final result. $\quad\square$

**The proof of R-SVRG (Theorem 5.13 and Corollary 5.14)**

**Lemma B.4.** *(Lemma 1 in (Han & Gao, 2021b)) Suppose Assumptions 2.2 and 2.3 hold and consider the estimator of RSVRG. We have*

$$\mathbb{E}\left[\|\Delta_k^s\|^2\right] \leqslant \frac{k}{b'}\tilde{L}^2\mu^2\nu^2\eta^2\sum_{i=0}^{k-1}\mathbb{E}\left[\|v_i^s\|^2\right] + \mathbb{1}_{\{b<n\}}\frac{\sigma^2}{b}.$$

*Proof.* By Lemma B.4, $\mathcal{M}_1 = \frac{\mathcal{L}}{b'}$, $\gamma_{k,i} = k$, $\mathcal{M}_2 = \frac{\mathbb{1}_{\{b<n\}}}{b}$, $\lambda_k = 1$. Thus we have $\hat{\Gamma} = m^2$, $\hat{\Lambda} = Sm$, $\varphi_5 = 2\left(\frac{\eta^2 m^2 \mathcal{L}}{b'}+1\right)$, $\phi_5 = \eta\left(\frac{1-\eta L}{2} - \frac{\eta^2 m^2 \mathcal{L}}{2b'}\right)$. According to Theorem 5.1, we obtain

$$\mathbb{E}\left[\|\operatorname{grad} f(x_\zeta)\|^2\right] \leqslant \frac{\varphi_5 \tilde{\Delta}_0}{\phi_5 K} + \frac{(\varphi_5 \eta + 4\phi_5)\mathbb{1}_{\{b<n\}}\sigma^2}{2\phi_5 b}.$$

Consider the parameter setting $m = \sqrt{b'}$ and $\eta = \min\left\{\frac{1}{2L}, \frac{1}{2\tilde{L}\mu\nu}\right\}$, then we have $\mathbb{E}\left[\|\operatorname{grad} f(x_\zeta)\|^2\right] \leqslant \frac{20\tilde{\Delta}_0}{\eta K} + \frac{18\sigma^2}{b}$. In finite-sum setting, if we choose $b = n$ and $b' = n^{2/3}$, then we require that $K = \Theta\left(\varepsilon^{-2}\right)$ to obtain $\varepsilon$-accurate solution. Consequently, the total IFO complexity is given by $S(b + 2mb') = Sn + 2Kb' = \mathcal{O}\left(\frac{n^{2/3}}{\varepsilon^2}\right)$. Under online case, choosing $b = \Theta\left(\varepsilon^{-2}\right)$ and $b' = \Theta\left(\varepsilon^{-4/3}\right)$, then we require $K = \Theta\left(\varepsilon^{-2}\right)$. Therefore, the total IFO complexity to obtain $\varepsilon$-accurate solution is $S(b + 2mb') = \frac{1}{\sqrt{b'}\varepsilon^2}(b + 2mb') = \mathcal{O}\left(\varepsilon^{-10/3}\right)$. □

**The proof of R-PAGE (Theorem 5.16 and Corollary 5.17)**

We first present a necessary lemma.

**Lemma B.5.** *Suppose Assumptions 2.2 and 2.3 hold. If the estimator $v_k$ is defined as (10), then we have*

$$\mathbb{E}\left[\|\Delta_k\|^2\right] \leqslant \frac{\tilde{L}^2\eta^2}{b'}\sum_{i=0}^{k-1}(1-p)^{k-i}\mathbb{E}\left[\|v_i\|^2\right] + \mathbb{1}_{\{b<n\}}\frac{\sigma^2}{b}.$$

*Setting $b = n$ in finite-sum settings, we have*

$$\mathbb{E}\left[\|\Delta_k\|^2\right] \leqslant \frac{\tilde{L}^2\eta^2}{b'}\sum_{i=0}^{k-1}(1-p)^{k-i}\mathbb{E}\left[\|v_i\|^2\right].$$

*Proof.* Denote $\Delta_k = v_k - \operatorname{grad} f(x_k)$. By the definition of $v_k$ in (10), we have

$\mathbb{E}\left[\|\Delta_k\|^2\right]$

$= p\mathbb{E}\left[\left\|\frac{1}{b}\sum_{i\in\mathcal{B}}\operatorname{grad} f_i(x_k) - \operatorname{grad} f(x_k)\right\|^2\right]$

$\quad + (1-p)\mathbb{E}\left[\left\|\mathcal{T}_{x_{k-1}}^{x_k}v_{k-1} + \frac{1}{b'}\sum_{i\in\mathcal{B}'}\left(\operatorname{grad} f_i(x_k) - \mathcal{T}_{x_{k-1}}^{x_k}\operatorname{grad} f_i(x_{k-1})\right) - \operatorname{grad} f(x_k)\right\|^2\right]$

$= (1-p)\mathbb{E}\left[\left\|\mathcal{T}_{x_{k-1}}^{x_k}v_{k-1} + \frac{1}{b'}\sum_{i\in\mathcal{B}'}\left(\operatorname{grad} f_i(x_k) - \mathcal{T}_{x_{k-1}}^{x_k}\operatorname{grad} f_i(x_{k-1})\right) - \operatorname{grad} f(x_k)\right\|^2\right] + \mathbb{1}_{\{b<n\}}\frac{p\sigma^2}{b}$

$= (1-p)\mathbb{E}\left[\left\|\mathcal{T}_{x_{k-1}}^{x_k}\Delta_{k-1} + \frac{1}{b'}\sum_{i\in\mathcal{B}'}\left(\operatorname{grad} f_i(x_k) - \mathcal{T}_{x_{k-1}}^{x_k}\operatorname{grad} f_i(x_{k-1})\right) + \mathcal{T}_{x_{k-1}}^{x_k}\operatorname{grad} f(x_{k-1}) - \operatorname{grad} f(x_k)\right\|^2\right]$

$\quad + \mathbb{1}_{\{b<n\}}\frac{p\sigma^2}{b}$

$$= (1-p)\,\mathbb{E}\left[\|\Delta_{k-1}\|^2\right] + \mathbb{1}_{\{b<n\}}\frac{p\sigma^2}{b}$$

$$+ (1-p)\,\mathbb{E}\left[\left\|\frac{1}{b'}\sum_{i\in\mathcal{B}'}\left(\operatorname{grad} f_i(x_k) - \mathcal{T}^{x_k}_{x_{k-1}}\operatorname{grad} f_i(x_{k-1})\right) + \mathcal{T}^{x_k}_{x_{k-1}}\operatorname{grad} f(x_{k-1}) - \operatorname{grad} f(x_k)\right\|^2\right]$$

$$\leqslant (1-p)\,\mathbb{E}\left[\|\Delta_{k-1}\|^2\right] + \frac{(1-p)}{b'}\mathbb{E}\left[\left\|\operatorname{grad} f_i(x_k) - \mathcal{T}^{x_k}_{x_{k-1}}\operatorname{grad} f_i(x_{k-1})\right\|^2\right] + \mathbb{1}_{\{b<n\}}\frac{p\sigma^2}{b}$$

$$\leqslant (1-p)\,\mathbb{E}\left[\|\Delta_{k-1}\|^2\right] + \frac{(1-p)\tilde{L}^2\eta^2}{b'}\mathbb{E}\left[\|v_{k-1}\|^2\right] + \mathbb{1}_{\{b<n\}}\frac{p\sigma^2}{b},$$

where the second equality is due to Assumption 2.2 (3), the first inequality holds since $\mathbb{E}[\|x - \mathbb{E}[x]\|^2] \leqslant \mathbb{E}[\|x\|^2]$. By using the Lemma B.1, the last inequality holds. Recursively applying this inequality gives

$$\mathbb{E}\left[\|\Delta_k\|^2\right] \leqslant \frac{\tilde{L}^2\eta^2}{b'}\sum_{i=0}^{k-1}(1-p)^{k-i}\mathbb{E}\left[\|v_i\|^2\right] + \mathbb{1}_{\{b<n\}}\frac{p\sigma^2}{b}\sum_{i=0}^{k-1}(1-p)^{k-i-1}$$

$$\leqslant \frac{\tilde{L}^2\eta^2}{b'}\sum_{i=0}^{k-1}(1-p)^{k-i}\mathbb{E}\left[\|v_i\|^2\right] + \mathbb{1}_{\{b<n\}}\frac{\sigma^2}{b}.$$

In finite-sum settings, if we choose $b = n$, then we have

$$\mathbb{E}\left[\|\Delta_k\|^2\right] \leqslant \frac{\tilde{L}^2\eta^2}{b'}\sum_{i=0}^{k-1}(1-p)^{k-i}\mathbb{E}\left[\|v_i\|^2\right].$$

$\square$

### The proof of Theorem 5.16 and Corollary 5.17

*Proof.* By Lemma B.5, we have $\mathcal{M}_1 = \frac{\tilde{L}^2}{b'}$, $\mathcal{M}_2 = \frac{\mathbb{1}_{\{b<n\}}}{b}$, $\gamma_{k,i} = (1-p)^{k-i}$, $\lambda_k = 1$. Thus

$$\hat{\Gamma} = \frac{1}{p} \geqslant \sum_{k=0}^{m-1}\gamma_{k,0}, \quad \hat{\Lambda} = K = Sm \geqslant \sum_{s=1}^{S}\sum_{k=0}^{m-1}\lambda_k.$$

According to Theorem 5.1, it is easy to obtain

$$\mathbb{E}\left[\|\operatorname{grad} f(x_\zeta)\|^2\right] \leqslant \frac{\varphi_6\tilde{\Delta}_0}{\phi_6 K} + \frac{\varphi_6\eta + 4\phi_6}{2\phi_6}\cdot\frac{\mathbb{1}_{\{b<n\}}\sigma^2}{b},$$

where $\varphi_6 = 2\left(\frac{\tilde{L}^2\eta^2}{b'p} + 1\right)$ and $\phi_6 = \eta\left(\frac{1-\eta L}{2} - \frac{\tilde{L}^2\eta^2}{2b'p}\right)$. Set $p = \frac{b'}{b+b'}$, $b' \leqslant \sqrt{b}$ and $\eta \leqslant \min\left\{\frac{b'}{2\tilde{L}\sqrt{b+b'}}, \frac{1}{2L}\right\}$, then $2 \leqslant \varphi_6 \leqslant \frac{5}{2}$, $\frac{\eta}{8} \leqslant \phi_6 \leqslant \frac{\eta}{2}$.

In finite-sum setting, if we set $b = n$, then we have $\mathbb{E}\left[\|\operatorname{grad} f(x_\zeta)\|^2\right] \leqslant \frac{20\tilde{\Delta}_0}{\eta K}$. To obtain $\varepsilon$-accurate solution, it is necessary to satisfy the condition $\frac{20\tilde{\Delta}_0}{\eta K} \leqslant \varepsilon^2$, which implies that the number of iterations $K$ must be at least $\frac{20\tilde{\Delta}_0}{\eta\varepsilon^2}$. Consequently, the total IFO complexity is $b + K(pb + (1-p)b') = b + \frac{2n}{\varepsilon^2\sqrt{n+b'}} \sim \mathcal{O}\left(n + \frac{\sqrt{n}}{\varepsilon^2}\right)$.

Under online setting, we obtain $\mathbb{E}\left[\|\operatorname{grad} f(x_\zeta)\|^2\right] \leqslant \frac{20\tilde{\Delta}_0}{\eta K} + \frac{18\sigma^2}{b}$. To obtain $\varepsilon$-accurate solution, we require $\frac{20\tilde{\Delta}_0}{\eta K} \leqslant \frac{\varepsilon^2}{2}$ and $\frac{18\sigma^2}{b} \leqslant \frac{\varepsilon^2}{2}$. Thus $K = \Theta(\frac{\sqrt{b+b'}}{b'\varepsilon^2})$ and $b = \Theta(\varepsilon^{-2})$. Consequently, the total IFO complexity is $b + K(pb + (1-p)b') = b + \frac{2b'bK}{b+b'} = \mathcal{O}\left(\frac{1}{\varepsilon^2} + \frac{1}{\varepsilon^4\sqrt{b+b'}}\right) = \mathcal{O}\left(\frac{1}{\varepsilon^2} + \frac{1}{\varepsilon^3}\right)$. $\square$

# C. Experiment Details and Additional Experiment Results

## C.1. Geometry of Specific Riemannian Manifolds

### C.1.1. GRASSMANN MANIFOLD

The Grassmann manifold $\mathrm{Gr}(r, d)$ has the structure of a Riemannian quotient manifold (Absil et al., 2008), i.e., $\mathrm{Gr}(r, d) :=$ $\mathrm{St}(r, d)/O(r)$, where $\mathrm{St}(r, d)$ is the Stiefel manifold, which is the set of matrices of size $d \times r$ with orthonormal columns. An element of the Grassmann manifold is represented by a $d \times r$ orthogonal matrix $\mathbf{U}$ with orthonormal columns, satisfying the condition $\mathbf{U}^\top \mathbf{U} = \mathbf{I}$. Any element is considered equivalent to $\mathbf{U}$ if it can be expressed as $\mathbf{UR}$ for any $\mathbf{R} \in O(r)$. We use the polar-based retraction $R_\mathbf{X}$, which is commonly used for Riemannian optimization (Zhou et al., 2021; Han & Gao, 2021b; Boumal et al., 2014). Denote $\mathrm{p}f$ represents the extraction of polar factors from polar decomposition. Then $R_\mathbf{X}(\mathbf{V}) = \mathrm{p}f(\mathbf{X} + \mathbf{V})$, the inverse retraction is $R_\mathbf{X}^{-1}(\mathbf{Y}) = \mathbf{Y} \left(\mathbf{X}^\top \mathbf{Y}\right)^{-1} - \mathbf{X}$. The associated vector transport is the orthogonal projection to the horizontal space, i.e., $\mathcal{T}_\mathbf{X}^\mathbf{Y}(\mathbf{V}) = \left(\mathbf{I} - \mathbf{YY}^\top\right) \mathbf{V}$. Although the vector transport method is not isometric, it is feasible to create isometric vector transport using singular value decomposition (SVD), albeit at a high computational cost. Nevertheless, Han & Gao (2021b) demonstrated that non-isometric vector transport performed effectively.

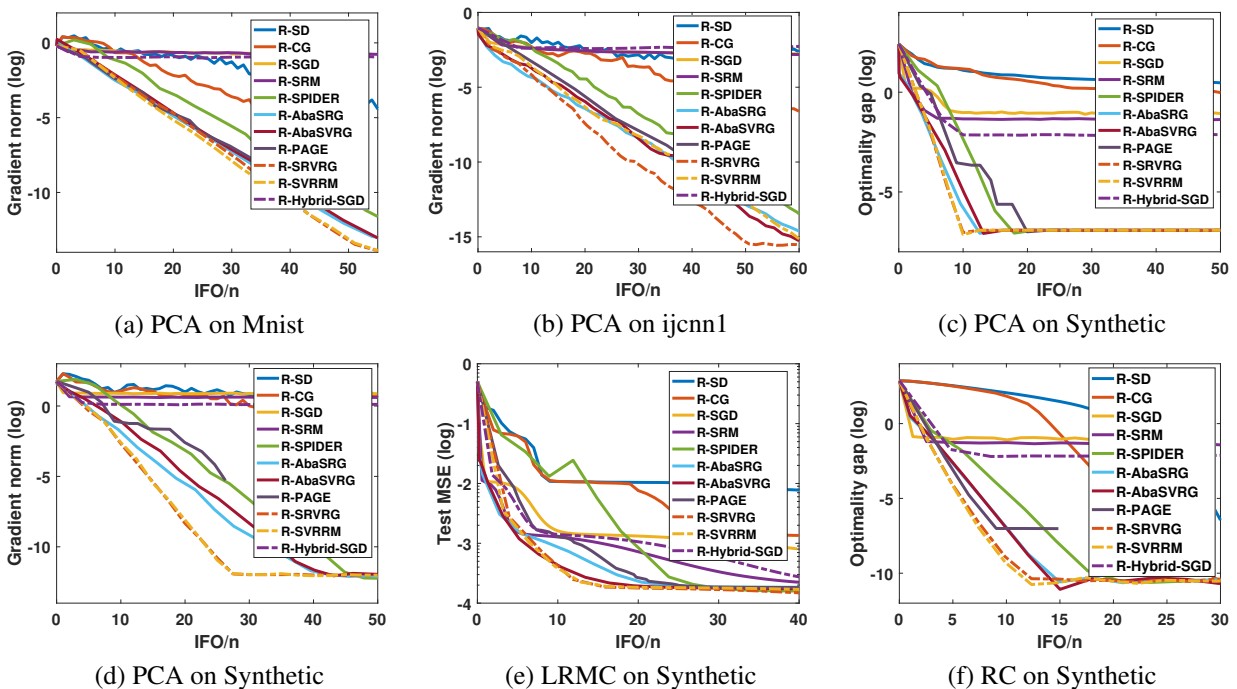

(a) PCA on Mnist     (b) PCA on ijcnn1     (c) PCA on Synthetic

(d) PCA on Synthetic     (e) LRMC on Synthetic     (f) RC on Synthetic

*Figure 2.* Convergence results for the PCA problem in terms of gradient norm on Mnist and ijcnn1, along with additional results on synthetic dataset.

### C.1.2. SYMMETRIC POSITIVE DEFINITE (SPD) MANIFOLD

The symmetric positive definite (SPD) manifold $\mathcal{S}_{++}^d$ is defined as the set of $d \times d$ symmetric positive definite matrices, represented as $\mathbb{S}_{++}^d := \{\mathbf{X} \in \mathbb{R}^{d \times d} : \mathbf{X}^\top = \mathbf{X}, \mathbf{X} \succ 0\}$. By endowing $\mathcal{S}_{++}^d$ with the affine-invariant Riemannian metric (AIRM) (Pennec et al., 2006) defined as $\langle \mathbf{U}, \mathbf{V} \rangle_\mathbf{X} = \mathrm{tr}\left(\mathbf{X}^{-1}\mathbf{U}\mathbf{X}^{-1}\mathbf{V}\right)$ for $\mathbf{U}, \mathbf{V} \in \mathrm{T}_\mathbf{X}\mathcal{S}_{++}^d$ at $\mathbf{X} \in \mathcal{S}_{++}^d$, the SPD manifold $\mathcal{S}_{++}^d$ forms a Riemannian manifold. We adopt the retraction $R_\mathbf{X}(\mathbf{V}) = \mathbf{X} + \mathbf{V} + \frac{1}{2}\mathbf{V}\mathbf{X}^{-1}\mathbf{V}$ from (Kasai et al., 2018b; Han & Gao, 2021b), along with the isometric vector transport. Note that the isometric vector transport can be derived by parallelization for $\mathcal{S}_{++}^d$. That is, For any $\mathbf{X}, \mathbf{Y} \in \mathcal{S}_{++}^d$, the expression $\mathcal{T}_\mathbf{X}^\mathbf{Y}\xi = \mathbf{B}_\mathbf{Y}\mathbf{B}_\mathbf{X}^\mathrm{b}\xi$ holds, where $\mathbf{B}_\mathbf{X} \in \mathbb{R}^{d \times d}$ represents the orthonormal basis on $\mathrm{T}_\mathbf{X}\mathcal{S}_{++}^d$ and $\mathbf{B}_\mathbf{X}^\mathrm{b} : \mathrm{T}_\mathbf{X}\mathcal{S}_{++}^d \to \mathbb{R}$ is defined such that $\mathbf{B}_\mathbf{X}^\mathrm{b}\mathbf{U} = \langle \mathbf{B}_\mathbf{X}, \mathbf{U} \rangle_\mathbf{X}$.

## C.2. Parameter Settings and Additional Experiment Results

For the probability $p$ of PAGE in all experiments, we select from $\{\frac{19}{20}, \frac{39}{40}, \frac{79}{80}, \frac{159}{160}, \frac{319}{320}\}$, the mini-batch size is set to be $b = n$, $b' = \sqrt{n}$. For R-Hybrid-SGD, we set $\beta = 0.01$ across all experiments.

### C.2.1. PCA ON GRASSMANN MANIFOLD

For all double-loop algorithms, we set $\eta = 0.03$ on MNIST [2] and $\eta = 0.9$ on ijcnn1[3]. The step sizes for R-SRM and R-Hybrid-SGD are set to 0.05 and 0.3 on datasets MNIST and ijcnn1, respectively. In the experiments conducted on Mnist and ijcnn1, our proposed algorithm selected parameters $\beta = 0.1$ and $\beta = 0.5$, respectively. In fact, the algorithm exhibits insensitivity to the choice of $\beta$. Figure 2 (a) and (b) illustrate the comparison of gradient norm descent when the algorithms solve the PCA problem on the MNIST and ijcnn1 datasets, respectively. The results demonstrate that our proposed R-SRVRG and R-SVRRM perform more effectively. Figure 2 (c) and (d) present the comparison of the performance on the synthetic dataset with $(n, d, r) = (10^5, 200, 5)$ in terms of optimality gap and gradient norm, where the synthetic data is sourced from (Han & Gao, 2021b). Similar observations can be made.

### C.2.2. LRMC ON GRASSMANN MANIFOLD

We set $\eta = 7 \times 10^{-5}$ on Movielens-1M[4] and $\eta = 9 \times 10^{-6}$ on Jester[5] for both R-SRVRG and R-SVRRM. The step sizes for R-SRM and R-Hybrid-SGD are set to $5 \times 10^{-5}$ and $5 \times 10^{-6}$, respectively, on Movielens-1M with 6040 users ($d$) and 3706 movies ($n$) (Harper & Konstan, 2015) and Jester (Goldberg et al., 2001) with 24983 users and 100 jokes. In the experiments conducted on the Movielens-1M and Jester datasets, our proposed algorithm utilized parameter values of $\beta = 0.5$ and $\beta = 0.1$, respectively. Figure 2 (e) demonstrates the performance of solving the LRMC problem on synthetic data with $(n, d, r) = (20000, 100, 5)$ (sourced from (Han & Gao, 2021b)). It can be observed that our R-SRVRG and R-SVRRM exhibit superior performance.

### C.2.3. RC ON SPD MANIFOLD

For the experiments conducted on the Extended Yale B dataset, we set $\eta = 0.03$ for all VR methods and $\eta = 0.04$ for both R-SRM and R-Hybrid-SGD. For Kylberg[6] dataset, we set $\eta = 0.04$ for R-AbaSRG and R-AbaSVRG methods, and choose $\eta = 0.05$ for the other methods. We set $\beta = 0.5$ for our proposed algorithms in the experiments conducted on both Yale B and Kylberg datasets. In Figure 2 (f), we compare the performance of solving the RC problem on synthetic data with $(n, d, cn) = (5000, 100, 5)$[7]. Our proposed R-SRVRG and R-SVRRM continue to demonstrate excellent performance.

## C.3. Sensitivity of parameter $\beta$

We demonstrate the robustness of parameter $\beta$ in the proposed R-SRVRG, R-SVRRM, and R-Hybrid-SGD algorithms, respectively. Figure 3 illustrates the effect of parameter $\beta$ on the R-SRVRG algorithm across different tasks and datasets, indicating that the algorithm is robust to the selection of $\beta$. In Figures 4 and 5, we present the effects of parameter $\beta$ on both the R-SVRRM algorithm and the R-Hybrid-SGD algorithm, indicating that our proposed algorithms exhibit insensitivity to the selection of $\beta$.

---

[2]http://yann.lecun.com/exdb/mnist/
[3]https://www.csie.ntu.edu.tw/~cjlin/libsvmtools/datasets/
[4]https://grouplens.org/datasets/movielens/
[5]https://eigentaste.berkeley.edu/
[6]https://www.cb.uu.se/~gustaf/texture/
[7]source: https://github.com/andyjm3/R-AbaVR

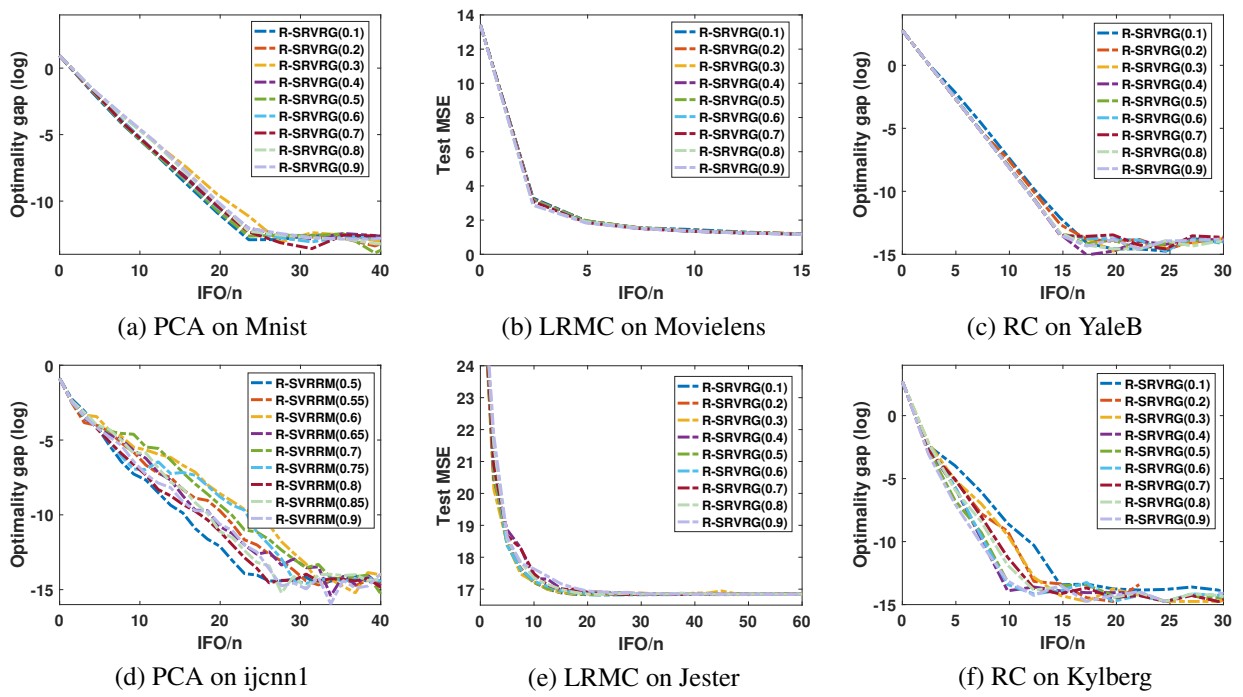

Figure 3. Robustness of parameter $\beta$ in the R-SRVRG algorithm.

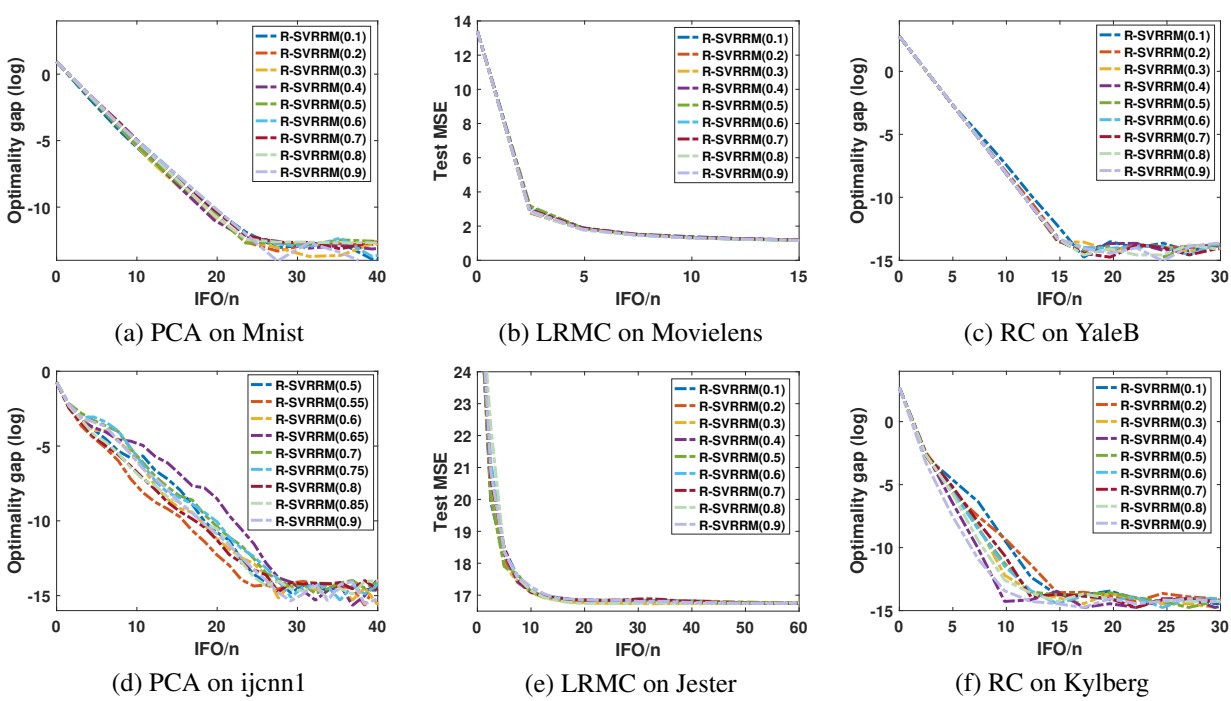

Figure 4. Robustness of parameter $\beta$ in the R-SVRRM algorithm.

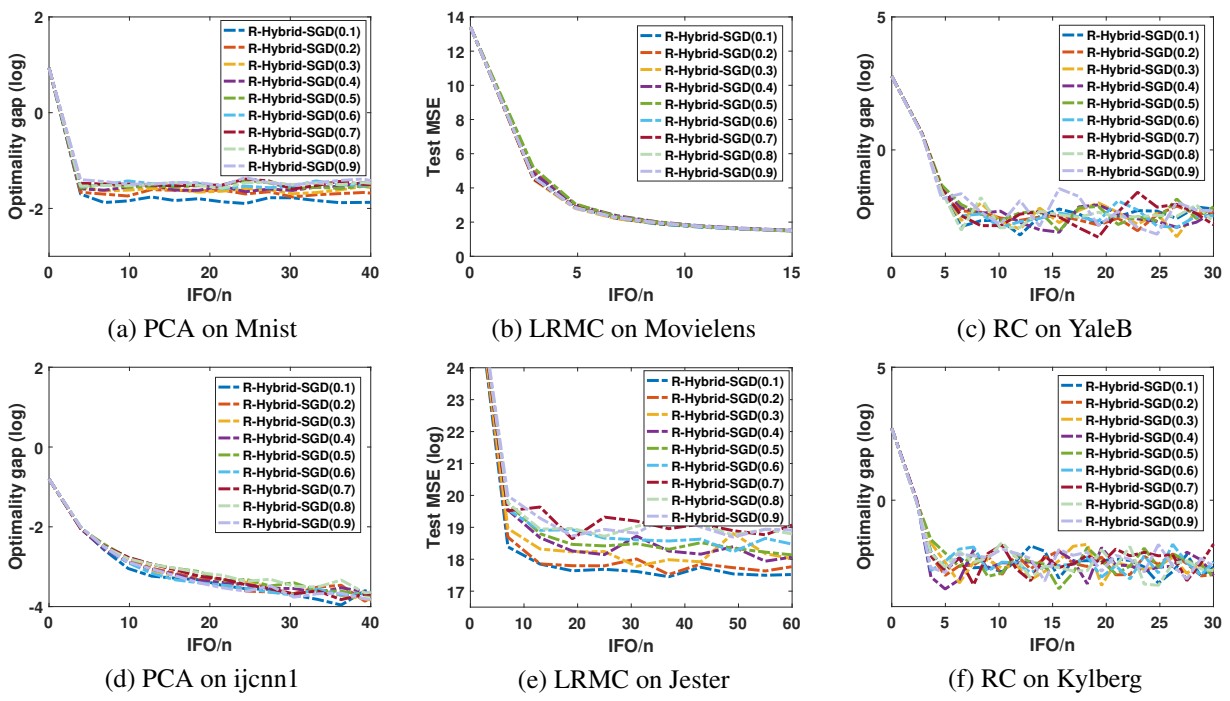

*Figure 5.* Robustness of parameter $\beta$ in the R-Hybrid-SGD algorithm.

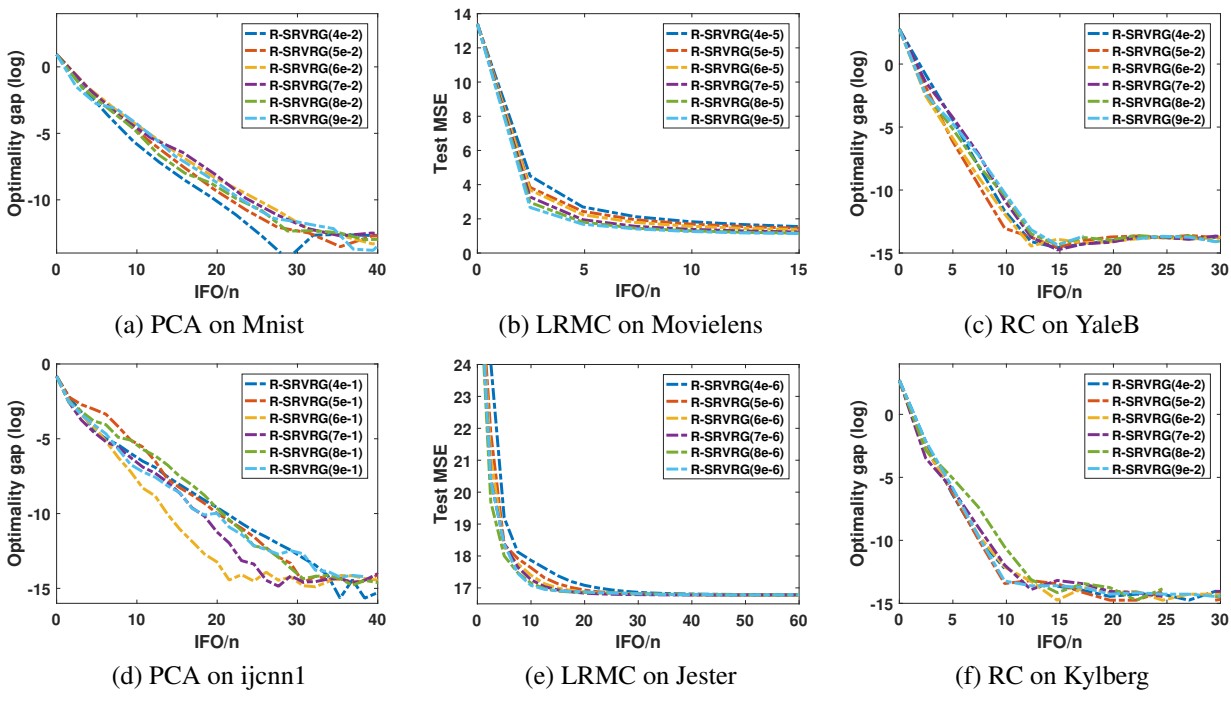

*Figure 6.* Robustness of step sizes $\eta$ in the R-Hybrid-SGD algorithm.

## C.4. Sensitivity of step sizes $\eta$

Figures 6, 7, and 8 illustrate the effects of different constant step sizes on our proposed algorithms. As observed in Figures 6 and 7, the selection of step size is relatively insensitive for the R-SRVRG and R-SVRRM algorithms. Figure 8 shows a

somewhat greater impact of step size on the R-Hybrid-SGD algorithm, which also indicates that our improved algorithm, R-SRVRG, enhances the robustness of the method.

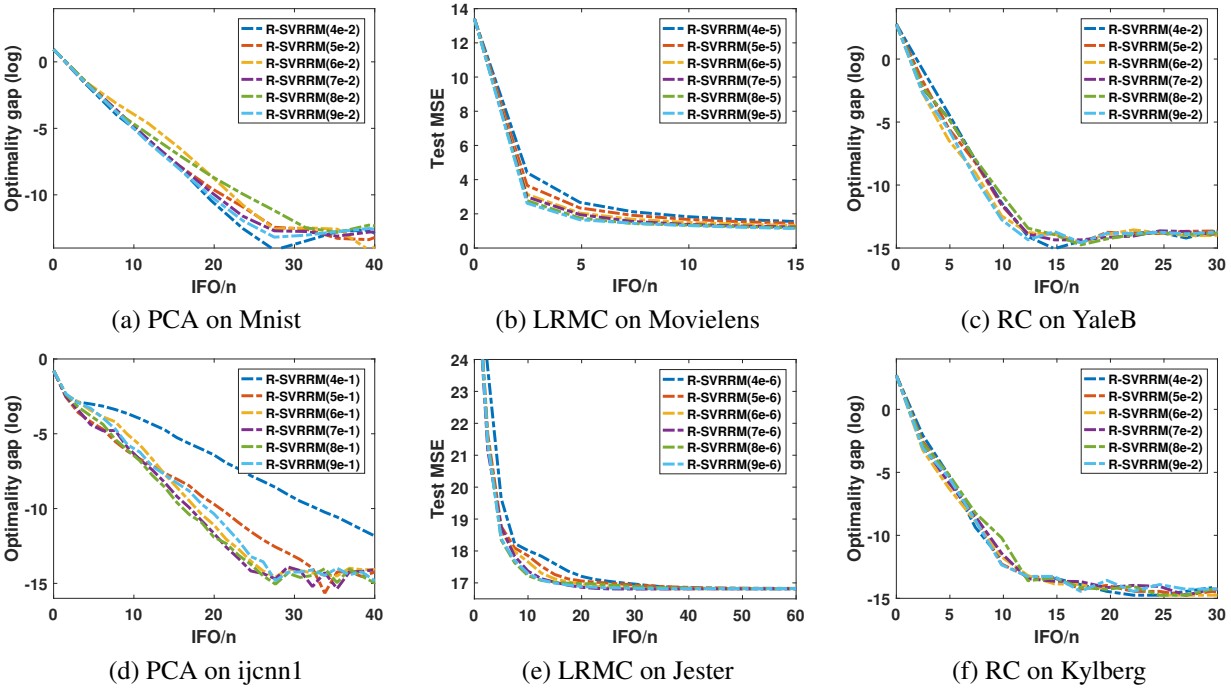

Figure 7. Robustness of step sizes $\eta$ in the R-Hybrid-SGD algorithm.

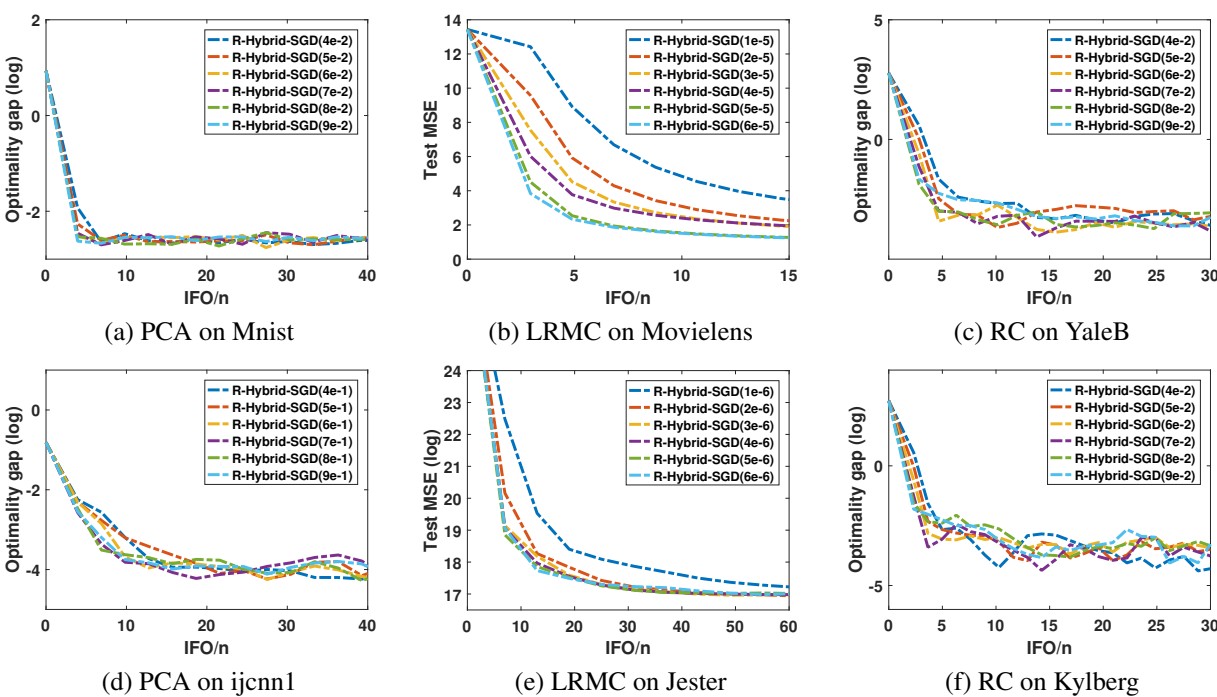

Figure 8. Robustness of step sizes $\eta$ in the R-Hybrid-SGD algorithm.

