# OpenReview forum: "SERENA: A Unified Stochastic Recursive Variance Reduced Gradient Framework for Riemannian Non-Convex Optimization"
_ICML.cc/2025/Conference — ICML 2025 poster_

### Official Review · Reviewer_PqwF · 2025-02-18

**Overall Recommendation:** 3

**Summary:**

The paper presents a variance reduction framework for Riemannian non-convex optimization. The proposed framework covers many existing variance reduction algorithms on manifolds and theoretical analysis matches the best known results in the Euclidean space. Numerical experiments are conducted on PCA, LRMC and RC problems and benchmarked against extensive baselines. The performance of the proposed algorithm seems promising.

**Claims And Evidence:**

It is a bit unclear what is the extent of the contributions of this paper. The paper seems to propose a unified framework (based on hybrid-SGD, a method already proposed in the Euclidean space). In this regard, the contribution seems limited, given the analysis would readily follow from the paper. On the other hand, it seems the paper also establishes a recursive variance reduction framework, which seems novel, provided no such algorithms exist in the Euclidean space.

**Essential References Not Discussed:**

All the key related works have been cited.

**Experimental Designs Or Analyses:**

The experiment designs are standard. However, the analyses of the experiment results lack depth and insights. For example, how does the change in beta improves the convergence.

**Methods And Evaluation Criteria:**

The evaluation criteria is standard for the problem and the datasets/problem instances are commonly used as benchmarks for Riemannian non-convex optimization.

**Other Comments Or Suggestions:**

(1) the presentation of the results can be improved. The constants in the theorems can be simplified.

(2) Assumption 2.2(2), Riemannian Hessian bounded not written out explicitly.

(3) There are too many assumptions and it is unclear when such assumptions are satisfied.

**Other Strengths And Weaknesses:**

It is unclear about the contributions of this work, relative to existing works in the Euclidean space.

**Questions For Authors:**

Please clarify the contributions relative to the prior variance reduction methods in the Euclidean space.

**Relation To Broader Scientific Literature:**

Established a unified framework, which completes the variance reduction methods for Riemannian optimization.

**Theoretical Claims:**

I briefly went through the proof in the appendix. The proof seems to be correct.

---

> ### Author Rebuttal · Authors · 2025-03-29
>
> Thank you for your valuable comments. You are primarily concerned with the contributions of this paper compared to previous variance reduction algorithms in Euclidean space, as well as whether the assumptions made in this paper can be satisfied. Additionally, you are interested in the impact of the parameter  $\beta$ on convergence and the simplification of symbols in the theorem. Below, we will address each of these points in detail.
> 1. Compared to previous work in Euclidean space, our contributions are as follows.
>
> (1) Firstly, we propose the SRVRG estimator, which does not exist in Euclidean space. Secondly, we present a unified theoretical analysis for the first time.  In non-convex settings, the theoretical analysis of various variance reduction algorithms shows significant differences even in Euclidean space, with some requiring the construction of Lyapunov functions.
>
> (2) Our Corollary 5.4 indicates that SRVRG exhibits superiority in parameter selection. For instance, in finite-sum case where $ n < \varepsilon^{-2} $, when the other parameter settings adhere to those specified in the Corollary, it is sufficient for $ \beta $ and $ b'$ to satisfy ${\beta}/{b'} = n^{-1} $ to achieve optimal complexity. Given that $ \beta = m^{-1} $, it follows that $ mb' = n $. For other algorithms, although some can also achieve optimal complexity, the parameters in their analyses are fixed; for example, in R-AbaSRG, $ b' = m = n^{1/2}$. Our experimental results also validate the robustness of $\beta$. We will add a remark after Corollary 5.4 to clarify the following point, for example, "Corollary5.4 indicates that if $ \beta $ and $ b'$ satisfy $\sqrt {\frac{\beta }{{b'}}}  = \max \\{ {\frac{1}{{\sqrt n }},\varepsilon } \\}$ (finite-sum case) or $\sqrt {\frac{\beta }{{b'}}}  =  {\varepsilon }$ (online case), our proposed algorithm can achieve optimal complexity. This highlights the advantages of our algorithm in terms of parameter selection.".
>
> 2. Due to space limitations, we omitted the explanation of the assumptions outlined below.
>
> (1) Assumption 2.2 (1) is to avoid a bad case in Riemannian optimization. Namely, the sequence $\{x_k\}$ may converge to an optimum $x_*$, while the connecting retraction $\{R_{x_k}(\xi_k)\}$ does not converge where $x_{k+1}=R_{x_k}(\xi_k)$. This assumption is also employed in papers (Zhou et al., 2021, Han \& Gao, 2021b, Kasai et al., 2018).
>
> (2) In Assumption 2.2 (2), the boundedness of the Riemannian Hessian is unnecessary, and we will remove it.
> Assumptions 2.2 (1) to (3) clearly hold for compact manifolds, including sphere, (compact) Stiefel and Grassmann manifolds. For non-compact manifolds, like SPD matrices, the assumptions hold by choosing a sufficiently small neighbourhood $\mathcal{X}$.
>
> (3) Assumption 2.2 (4) generalizes the notion of smoothness on the euclidean space to Riemannian manifold and is satisfied if $\frac{d^2 f(R_x(t \xi))}{d t^2} \leq L$ for all $x \in \mathcal{X}, \xi \in T_x \mathcal{M}$ with $\||\xi\||=1$.
> Assumption 2.2 (5) and (6) are necessary due to our reliance on vector transport to approximate parallel transport. These two assumptions can be derived by requiring the vector transport $\mathcal{T}$ to be isometric and satisfy $\|| \mathcal{T}\_{x}^y u- \mathrm{D}R_x(\xi)[u] \|| \leq c_0 \|| \xi \|| \|| u \||$, where $\mathrm{D}R_x(\xi)[u]=\frac{d}{d t} R_x(\xi+t u)|_{t=0}$ is the differentiated retraction. Assumption 2.2 (6) is ensured by Taylor approximation in a compact set $\mathcal{X}$.
>
> (4) Assumption 2.3 (1) ensures that the Riemannian distance can be expressed in terms of the inverse exponential map. Assumption 2.3 (2) hence relates the exponential map with the retraction. Indeed, we have $\left\|R_x^{-1}(y)\right\| \leq \mu\left\|\operatorname{Exp}_x^{-1}(y)\right\|$ and $\left\|\operatorname{Exp}_x^{-1}(y)\right\| \leq \nu\left\|R_x^{-1}(y)\right\|$. This assumption remains valid when $\mathcal{X}$ is sufficiently small. These assumptions are also standard as in (Sato et al., 2019, Han \& Gao, 2021b)
>
> 3. The experimental results in the appendix C.3 indicate that the algorithm is robust to $\beta$. Theoretically, Corollary 5.4 indicates that setting $\sqrt{\beta/b'} = \varepsilon$ can ensure the optimal complexity of the algorithm, which means that $\beta$ and $b'$ can balance each other, and when a larger $b'$ is chosen, the algorithm will be more robust  to $\beta$.
>
> 4. We will simplify the symbols in the theorem, such as $\mathcal{L} = {{\tilde L}^2}{\mu ^2}{\nu ^2}$.

---

> > ### Comment · Reviewer_PqwF · 2025-04-09
> >
> > I thank the authors for detailed responses. Most of my concerns have been addressed and I would like to increase my rating to 3.

---

> > > ### Author Response · Authors · 2025-04-09
> > >
> > > Thank you for your positive feedback regarding our response. We are delighted to learn that you feel our reply effectively addressed your concerns. We sincerely appreciate your support and encouragement.

---

### Official Review · Reviewer_1swq · 2025-03-09

**Overall Recommendation:** 2

**Summary:**

The paper proposes the combination of SVRG and SARAH for stochastic variance reduced non-convex optimization, and extends it to the Riemannian setting. Combinations of Riemannian stochastic methods with or without variance reduction, existing or proposed in this paper,  all can be subsumed into a unified update with mix parameter beta which was proved to converge at a rate matching existing IFO complexity lower bound in the finite-sum or online setting. Experimental results are provided to demonstrate the slight advantage of the proposed Riemannian methods over existing ones.

**Claims And Evidence:**

I feel that the motivation of coming up with the combination of SVRG and SARAH for stochastic variance reduction is missing. Given a lot of existing combinations, why does the proposed combination stand out? Although experimental results demonstrate the proposed method converges faster, it is just slightly faster and measured in terms of IFO. Since the update is more complicated than existing ones, I guess this method doesn't work well in practice, e.g., in terms of running time.

Thus, I feel the contribution mainly lies in theoretical investigation on the combination of SVRG and SARAH, and a unified perspective that integrate all existing combinations.

**Essential References Not Discussed:**

Literature work seems complete.

**Experimental Designs Or Analyses:**

yes. My concern on experimental design is that there are so many baseline methods. It's unclear what is a good way to set their hyper-parameters in order for a fair comparison with the proposed method. For example, in the first paragraph of Section 6, it says "We selected the same parameters for R-SRVRG and R-SVRRM. Similarly, we chose identical parameters for R-SRM and R-Hybrid-SGD for comparison". But why?

Also, as I mentioned previously, it might be a totally different case if the x-axis is running time.

**Methods And Evaluation Criteria:**

Evaluation of the methods looks reasonable.

**Other Comments Or Suggestions:**

In Section 2, about exponential map definition, the derivative of r(t) should be evaluate at 0.

The first condition of vector transport seems incorrect. Also, why two notations for it, T_x^z and T_{\xi}(y)?

**Other Strengths And Weaknesses:**

NULL

**Questions For Authors:**

see above

**Relation To Broader Scientific Literature:**

The paper proposes a new combination of R-SVRG and R-SARAH, and further provides a unified perspective on Riemannian stochastic variance reduced methods.

**Theoretical Claims:**

I didn't check proofs.

---

> ### Author Rebuttal · Authors · 2025-03-29
>
> We sincerely appreciate your valuable comments.
>
> 1. We will increase the discussion on the motivation behind the proposed SRVRG algorithm. The motivation for proposing SRVRG is that when $u_k$ is an SVRG-type estimator, parameter $\beta$ can be larger, which allows the variance of $v_k$ to decrease more rapidly (see the formula in the right column on lines 168-170 of the paper). Our theory also demonstrates that the range of $\beta$ is relatively expansive. Corollary 5.4 indicates that setting $\sqrt{\beta/b'} = \varepsilon$ or $\sqrt{\beta/b'} = n^{-1}$ can ensure the optimal complexity of R-SRVRG, which means that $\beta$ and $b'$ can balance each other. In contrast, Corollary 5.10 indicates that the R-SRM algorithm requires $\beta = K^{-2/3} = \mathcal{O}(\varepsilon^{2})$ to achieve optimal complexity. This is the reason why our algorithms perform better. Experimental results also validate the robustness of $\beta$.
>
> 2. Although the updates in our proposed algorithm are somewhat more complex, the average time for a single IFO call is 0.634s, which is 3.65 times the average time for a Riemannian update and 4.7 times the average time for a vector transport. This indicates that a significant portion of the algorithm's runtime is consumed by IFO calls, especially when a larger batch size $b'$ is selected in the inner loop. Our algorithm allows for a trade-off between the inner loop size $m$ and the batch size $b'$ (because in Corollary 5.4, $m = \beta^{-1}$), meaning that for the same $m$, our algorithm can support smaller batch sizes. Therefore, even when the x-axis represents running time, our algorithm still performs well. The comparison of algorithm performance with time on the x-axis demonstrates that our algorithm is comparable to the best-performing algorithms. See https://anonymous.4open.science/r/SERENA-RiemanOptimization/.
>
> 3. For the parameters of the other algorithms, we adopted most of the settings from (Han\&Gao,2021b). Additionally, we performed parameter optimization for certain algorithms, such as R-PAGE. The reason that algorithms R-SRVRG and R-SVRRM, as well as R-SRM and R-Hybrid-SGD, are selected with the same parameters is that algorithm R-SRVRG can cover R-SVRRM, and similarly, algorithm R-Hybrid-SGD can cover R-SRM.
>
> 4. Thank you very much for other comments or suggestions. The final part of the definition of the exponential mapping will be modified to $\dot{\gamma}(0)=\frac{d}{d t} \gamma(t)|_{t=0}=u$.  The first condition of vector transport will be modified to ''1) $\mathcal{T}$ has an associated retraction $R$, i.e., for $x \in \mathcal{M}$ and $\iota, \xi \in \mathrm{T}\_x \mathcal{M}, \mathcal{T}\_\xi(\iota)$ is a tangent vector at $R_x(\iota)$'' with the symbol $y$ is changed to $\iota$.

---

### Official Review · Reviewer_vLDf · 2025-03-13

**Overall Recommendation:** 4

**Summary:**

In this paper, the authors provide a unified framework for several Stochastic Recursive Variance Reduced Gradient algorithms. They first propose a new algorithm that integrates recursive momentum with variance reduction techniques, called Stochastic Recursive Variance Reduced Gradient (SRVRG), and extended it to Riemannian manifolds. Next, they propose a unified framework (SERENA), which includes SRVRG and extends to other variance reduction methods. Via studying their unified formulation, they derive Incremental First-order Oracle (IFO) complexity of finite-sum problems and online problems. These complexities are shown to match the theoretical lower bound for stochastic non-convex optimization.

**Claims And Evidence:**

Claims made in the submission are well supported by clear and convincing evidence.

**Essential References Not Discussed:**

I am not aware of any essential references not discussed.

**Experimental Designs Or Analyses:**

I did not check the soundness/validity of the experimental designs.

**Methods And Evaluation Criteria:**

The proposed framework makes sense as it unifies/generalizes a range of previous works.

**Other Comments Or Suggestions:**

I have no other comments or suggestions.

**Other Strengths And Weaknesses:**

Strengths:
- SERENA unifies several variance reduction techniques, achieving optimal IFO complexities for both finite-sum and online settings, improving upper bound results compared to prior analysis.
- The unified approach paves the way for further exploration of variance reduction techniques in manifold optimization.
- The formulation of recursive gradient estimators adapted to the geometry of Riemannian manifolds can influence future algorithm design.

Weaknesses:
- While the unification and improvements are valuable, the core ideas are extensions of well-established variance reduction methods in Euclidean spaces.

**Questions For Authors:**

I have no other questions for the authors.

**Relation To Broader Scientific Literature:**

The key contributions of the paper provides connections among several previous works, including SGD, SVRG-type, SARAH-type, Hybrid-SGD, STORM, SVRRM and the newly proposed SRVRG. (see Table 1 in the paper)

**Theoretical Claims:**

I did not check the correctness of proofs for theoretical claims.

---

> ### Author Rebuttal · Authors · 2025-03-29
>
> We thank the reviewer for the encouraging feedback and valuable comments.
>
> We introduce the SRVRG estimator, which is proposed for the first time in both Euclidean space and Riemannian space. Numerical experiments illustrate the superiority of the R-SRVRG algorithm. Furthermore, we present a unified framework for Riemannian stochastic variance reduction methods and provide a unified theoretical analysis by giving an upper bound on the variance of the Riemannian stochastic estimators. Our proposed algorithm can achieve optimal complexity while also offering a wide range of parameter choices. For example, when other parameter settings are as outlined in the Corollary 5.4, $\beta$ and $b'$ only need to satisfy $\sqrt {\frac{\beta }{{b'}}}  = \max \{ {\frac{1}{{\sqrt n }},\varepsilon } \}$ (finite-sum case) or $\sqrt {\frac{\beta }{{b'}}}  =  {\varepsilon }$ (online case) for the R-SRVRG algorithm to achieve optimal complexity. Other methods often directly specify parameters in order to achieve optimal complexity, such as in the analysis of R-AbaSRG for online case, $b' = \varepsilon^{-1}$. Furthermore, experimental results demonstrate robustness of parameters.

---

### Official Review · Reviewer_oiW7 · 2025-03-14

**Overall Recommendation:** 4

**Summary:**

This paper proposes a unified stochastic recursive variance reduced gradient framework for Riemannian non-convex optimization.

**Claims And Evidence:**

The algorithm and its analysis are presented with derivations and proofs, but clarification on assumptions and the construction of the variance reduced gradient needed.

**Essential References Not Discussed:**

N/A

**Experimental Designs Or Analyses:**

As commented earlier, it lacks a comparsion with Riemannian natural gradient methods.

**Methods And Evaluation Criteria:**

The applicability of the proposed methods is fine. But it lacks a comparison with SOTA Riemannian natural gradient methods, see,

Hu, J., Ao, R., So, A.M.C., Yang, M. and Wen, Z., 2024. Riemannian natural gradient methods. SIAM Journal on Scientific Computing, 46(1), pp.A204-A231.

**Other Comments Or Suggestions:**

- **Page 2, Line 57:** Typo – "euclidean" should be "Euclidean."

- **Assumption 2.2(1):** The requirement that all iterates stay within a totally retractive neighborhood of $x_*$ seems restrictive. Can this assumption be relaxed?

- **Assumption 2.2(2):** Is the assumption of a bounded Riemannian Hessian necessary for the analysis?

- **Equation (3):** What does $\tilde{x}$ represent? Please clarify.

- **Table 1:** Can you explain the meaning of each column for better clarity?

- **Second Line After Equation (7):** Should "SGD" be "SG"? It seems like it should refer to "stochastic gradient" rather than "stochastic gradient descent."

- **Algorithm 1:**
  - What are the sizes of $I_k$ and $J_k$?
  - How is $v_k^s$ computed? Note that Equation (7) provides the formula for $v_k$, not $v_k^s$.

- **Paragraph After Algorithm 1:**
  - The connections between R-SRM, Riemannian SVRG, and R-PAGE with Algorithm 1 are not clearly explained.
  - Additionally, some of these methods follow a single-loop structure, while Algorithm 1 appears to be a two-loop framework. Please clarify these distinctions.

- **Section 5:** The choice of \( S \) is missing in the convergence analysis. Please specify how it is determined.

**Other Strengths And Weaknesses:**

The optimal complexity for Riemannian variance-reduced methods, e.g., R-SRM (Han & Gao, 2021a), has already been established in prior work. This paper primarily unifies existing methods and follows standard proof techniques to recover the same complexity guarantees, offering limited new theoretical insights for the field.

**Questions For Authors:**

N/A

**Relation To Broader Scientific Literature:**

The paper proposes a unified variance-reduced gradient estimator on manifolds by consolidating existing methods from both Euclidean and Riemannian optimization literature.

**Theoretical Claims:**

The proofs seem correct.

---

> ### Author Rebuttal · Authors · 2025-03-30
>
> We appreciate your thorough review and useful comments. Please find our responses below.
>
> 1. Methods And Evaluation Criteria:
>
> Thank you for your valuable feedback regarding the applicability of our proposed methods. First, please allow me to explain why the RNGD method was not compared previously.
>
> (1) The RNGD algorithm utilizes an approximation of the Riemannian Hessian, making it a second-order method, while this paper primarily focuses on Riemannian stochastic variance reduction methods, which are the first-order methods.
>
> (2) This paper provide a unified framework and theoretical analysis, which cannot encompass the RNGD algorithm.
>
> We compared our proposed R-SRVGR and R-SVRRM with RNGD on the LRMC and PCA problems, respectively. The results indicate that the performance of our algorithms is comparable to that of RNGD. In particular, when the x-axis represents running time, ours perform better in most experiments. See details in https://anonymous.4open.science/r/SERENA-RieOptimization
>
> We will replace the last sentence in the conclusion and add references. "Furthermore, Riemannian second-order methods have garnered increasing attention,  such as the RNGD[1], R-SQN-VR[2], and R-SVRC[3] algorithms. We will explore the integration of this framework with second-order information to enhance the convergence rate.".
>
> 2. Other Strengths And Weaknesses:
>
> Although the optimal complexity of the Riemannian VR method has been established in previous work, R-SRVRG and R-SVRRM show better performance. Theoretically, we first provide a unified analytical framework, as the analyses of different algorithms previously varied significantly. Additionally, there are new theoretical insights; for instance,  when the other parameter settings adhere to those specified in the Corollary 5.4 in finite-sum case, our algorithms can guarantee optimal complexity as long as $mb'= \min\\{n,\varepsilon^{-2}\\}$ is satisfied and R-AbaSRG needs $ b' = m = n^{1/2}$. This indicates that our algorithms exhibits superiority in parameter selection, i.e., robustness and a broader range.
>
> 3. Other Comments Or Suggestions:
>
> (1) We will correct the typo.
>
> (2) Assumption 2.2(1) follows papers (Zhou et al., 2021, Han & Gao, 2021b, Kasai et al., 2018), and it is difficult to relax. In Assumption 2.2(2), the boundedness of the Riemannian Hessian is unnecessary, and we will remove it. More details for assumptions, see 2 in our rebuttal to Reviewer PqwF.
>
> (3) We will change $\tilde x$ to $\tilde x_{s-1}$ and provide the necessary explanations as follows " $\tilde x_{s-1}$ represents the point at which the true gradient is calculated in the outer loop of SVRG algorithm". For Equation (3) and (7), we will add superscript $s$ and indicate that it refers to the $s$-th outer loop. In second line after Equation (7), it is "SG".
>
> (4) We will add the caption for Table 1 as follows: Given $\beta$, the second and third columns represent the types of stochastic gradient estimators for $u$ and $w$, respectively, while the last column indicates the corresponding algorithms.
>
> (5) In Algorithm 1, $|\mathcal{I}_k| = |\mathcal{J}_k| = b'$, we will add " of size $b'$ " at the end of line 7. After Algorithm 1, we will add the following note "For single-loop algorithms, such as R-Hybrid-SGD and R-SRM, as well as loopless algorithms like R-PAGE, $S=1$ in Algorithm 1. And we omit the superscript for these algorithms to simplify notation, as in Equations (8) and (10).".
>
> (6) R-SRM, R-SVRG, and R-PAGE can be classified as special cases of Algorithm 1. These three algorithms are highlighted because they each exemplify distinct types: R-SRM has a single-loop structure with its stochastic estimate represented as a convex combination of SARAH-type and SG stochastic estimates. In contrast, R-SVRG employs a double-loop structure, while R-PAGE can be regarded as a loopless SARAH-type algorithm. We present these specific forms of the stochastic estimates to facilitate theoretical analysis.
>
> (7) For our R-SRVRG and R-SVRRM, $S = \frac{K}{m} = \Theta (\frac{1}{\sqrt {mb'} {\varepsilon ^2}})$. The optimal complexity can be guaranteed as long as $mb'= \min\\{n,\varepsilon^{-2}\\}$ is satisfied in finite sum case, thus $S = \Theta (\max \\{  n^{-1/2} {\varepsilon ^{-2},\frac{1}{\varepsilon }} \\})$. Similarly, in the online case, $mb'= \varepsilon^{-2}$, $S = \Theta ({\varepsilon ^{ - 1}})$. For R-SVRG, $S = \Theta(\varepsilon^{-2}n^{-1/3})$ in finite-sum case and $S= \Theta(\varepsilon^{-4/3})$.
> We will include the choice of $S$ in corollary.
>
> [1] Hu,J. et al. Riemannian natural gradient methods. SIAM J. Sci. Comput., 2024.
>
> [2] Kasai, H. et al. Riemannian stochastic quasi-newton algorithm with variance reduction and its convergence analysis. In AISTATS, 2018.
>
> [3] Zhang, D. and Davanloo Tajbakhsh, S. Riemannian stochastic variance-reduced cubic regularized newton method for submanifold optimization. J. Optim. Theory Appl., 2023.

---

> > ### Comment · Reviewer_oiW7 · 2025-04-02
> >
> > I appreciate the reviewer’s thoughtful responses and the addition of numerical tests, which address my concerns. I am happy to raise my rating.

---

> > > ### Author Response · Authors · 2025-04-03
> > >
> > > Thank you for your recognition of our responses and your positive feedback. We are pleased to hear that you believe the modifications we made and the additional numerical tests effectively address your concerns. Your support and encouragement are greatly appreciated.

---

### Official Review · Reviewer_JLi5 · 2025-03-24

**Overall Recommendation:** 5

**Summary:**

This paper proposes a generalization of stochastic recursive variance reduced gradient framework (SERENA) that unifies various gradient-like objects that serve as the first order information for Riemannian optimization. Based on the appropriate formulation of the gradient estimator proposed in this framework, the existing variance reduced algorithms on Riemannian optimization and the new Riemannian adaptations of existing Euclidean variance reduced algorithms are subsumed under the same framework which established a unified theoretical analysis on convergence. The convergence results of the newly proposed Riemannian adaptations, namely the Riemannian stochastic recursive variance reduced gradient algorithm (R-SRVRG), the Riemannian stochastic variance reduced momentum algorithm (R-SVRRM) and a Riemannian hybrid stochastic gradient descent algorithm (R-Hybird-SGD) are established, among which the R-SRVRG achieves the optimal incremental first-order oracle(IFO) complexity. The numerical experiments demonstrate superior performances of the proposed R-SRVRG and R-SVRRM in terms of IFO-complexity, i.e., the convergence with fewer stochastic gradient computations.

**Claims And Evidence:**

The claims made in this submission are accurately stated and supported by convincing evidence and discussions.

**Essential References Not Discussed:**

I am not aware of any other missing/not-cited related work that is essential to this paper.

**Experimental Designs Or Analyses:**

The experimental designs and analyses are solid to me. Other than the missing discussion on the impact of the Riemannian update, which was mentioned in the ``Methods And Evaluation Criteria'' section, I am a bit puzzled with the experiment on the Riemannian centroid. To the best of my knowledge, the Riemannian centroid (RC) problem on the SPD manifold with the affine-invariant metric is a convex optimization problem, as the SPD manifold with the affine-invariant metric is totally geodesically convex. How does that fit in this paper, which focuses on Riemannian non-convex optimization? While the convex optimization is certainly a more feasible setup than a non-convex one, does any experimental result pick out the implications brought by the convex condition with the RC?

**Methods And Evaluation Criteria:**

The proposed evaluation criteria, the use of the optimality gap/mean squared error along with IFO complexity as evaluation criteria is appropriate, as one of the compelling contributions of this paper is a unified theoretical analysis on IFO complexity. However, I would also expect the complexity discussion on the Riemannian update, retraction or exponential map.

While an analysis on IFO complexity alone is usually sufficient for a Euclidean stochastic first order algorithm with neglectable vector update, the update computation in a Riemannian setting is almost always a nontrivial primitive.  It is important to include the impact of the Riemannian update in the complexity analysis, especially when the ratios between the IFO call and update call are different. For example, R-SRVRG (Algorithm 2) makes $4b'$ calls to IFO computation and $1$ call to Riemannian update in the inner iteration, while R-SVRRM (Algorithm 3) makes $3b'$ calls to IFO computation and $1$ call to Riemannian update in the inner iteration. With that being said, a throughout complexity discussion that examines the profile of IFO computation and Riemannian update derails from the main focus and scope of this paper, as the profile significantly varies depending on the optimization problem, the Riemannian manifold and the choice of retraction map. I would prefer a simple discussion that reports the ratios between the IFO calls and update calls in different algorithms. It is recommended to also include the elapsed time (on average) for each call as a more practical reference.

**Other Comments Or Suggestions:**

Here is a list of minor things and suggestions.

1. As the paper mostly (if not always) works with the retraction, the discussion on manifolds and exponential mapping in the Preliminaries section 2 is too restrictive and unnecessary. It is not a good idea to mention a unique existence of geodesic globally:
	``If there exists a unique geodesic between any two points on $\mathcal{M},\cdots$''
	This is quite rare in practice (although the SPD manifold is one of those examples), so mentioning it might give the incorrect impression that the paper is limited to such cases.
1. The IFO complexity is not clearly defined. For those readers that are less familiar with the stochastic optimization like me, it could be difficult to relate IFO complexity with the counts of the call to stochastic gradient computation via Definition 2.1, especially when Def 2.1 says IFO complexity but it only defines what IFO is. The similar issue occurs to me in the Experiment section 6, which states that
``In Figure 1, the $x$-axis represents IFO(n)''.
Meanwhile, the figures says IFO/n and Def 2.1 says IFO takes an index and a point to return a gradient.
1. The $[n]$ in Def 2.1 is not defined before.
1. In section 3, check equation (2) and the equation above of a SARAH-type estimator. The $i_k$ and $j_k$ are not consistent with each other. Equation (2) should be Riemannian adaptation of that SARAH-type estimator, I tend to think this is a typo that will cause confusions, or correct me if there are reasons for that.
1. Appendix B.2.' title, ``The proof in Section 4'', I believe it should be Section 5.

**Other Strengths And Weaknesses:**

This submission is well written and organized. Readers can expect a fluent reading experience with it. The idea of proposing Riemannian adaptation of framework that unifies existing algorithms and derives new algorithms is always significant. As there are too many details in algorithmic designs vanish in the Euclidean setting, an appropriate adaptation that leads to one or more better Riemannian algorithms sheds some lights on the interpretations of the vanished algorithmic designs, which could have more implications beyond the scope of stochastic optimization.

**Questions For Authors:**

I do not have a question for the authors that would likely change my evaluation of the paper.

**Relation To Broader Scientific Literature:**

This paper propose a Riemannian framework (SERENA) for stochastic gradient estimators that adapts the existing Riemannian variance reduced algorithms including the Riemannian SRM, Riemannian SVRG and Riemannian PAGE. The R-SVRRM and R-SRVRG derived from SERENA not only outperforms (in terms of IFO complexity) the above-mentioned variance reduced algorithms, but also outperforms other existing Riemannian variance reduced algorithms that are not in SERENA. In addition, SERENA provides a unified theoretical analysis on IFO complexity for the algorithms it subsumes, which provides a different angle of interpretations for the existing algorithms.

**Theoretical Claims:**

Yes, I checked and confirmed the correctness of the proofs in appendix B. In particular, I read the proof of Theorem 5.3 and Corollary 5.4 in details.

---

> ### Author Rebuttal · Authors · 2025-03-29
>
> We thank the reviewer for the encouraging feedback and valuable comments and suggestions. We will respond to each point below.
>
> 1. Methods And Evaluation Criteria:
>
> Thank you for your insightful comments regarding the evaluation criteria and complexity analysis in our paper. As you mentioned, R-SRVRG makes $4b'$ calls to IFO computation and 1 call to Riemannian update in the inner iteration, while R-SVRRM makes $3b'$ calls to IFO computation and 1 call to Riemannian update in the inner iteration. In fact, for the R-SRVRG and R-SVRRM algorithms, the value of parameter $b'$ is closely related to $m$ (because Corollary 5.4 indicates that setting $\sqrt{\beta/b'} = \varepsilon$ or $\sqrt{\beta/b'} = n^{-1/2}$ can ensure the optimal complexity of R-SRVRG and R-SVRRM, and $\beta = m^{-1}$). When $m=n$ or $m=\varepsilon^{-2}$, $b'$ can be set to 1, the ratio $(b+ 4mb')/(m+1)= \Theta((n+\varepsilon^{-2})/m)$ between IFO calls and update calls can be a small constant $\mathcal{O}(1)$, our algorithm requires $S(1+m) = \Theta(\varepsilon^{-3})$ or $\Theta(n^{-1/2}\varepsilon^{-2})$ Riemannian updates. If we set $m = b'$, only $\Theta(\varepsilon^{-2})$ Riemannian updates are needed, but the ratio will be $\Theta(\varepsilon^{-1})$ or $\Theta(n^{1/2})$.
>
> From an experimental perspective, we recorded the time taken for 1 IFO call, 1 Riemannian update, and 1 vector transport across different problems and datasets. The average time for a single IFO call is 0.634s, which is 3.65 times the average time for a Riemannian update and 4.7 times the average time for a vector transport. This indicates that a significant portion of the algorithm's runtime is consumed by IFO calls, especially when a larger batch size is selected in the inner loop. Thus we can choose a smaller batch size $b'$.
>
> Furthermore, we also performed a comparison of the performance of various algorithms with the x-axis representing runtime. The results show that our proposed algorithm remains comparable to the best-performing algorithms. see https://anonymous.4open.science/r/SERENA-RiemanOptimization/.
>
> 2. Experimental Designs Or Analyses:
>
> In the experimental design, we selected the RC problem on the SPD manifold. As you mentioned, this problem is indeed a convex optimization problem. We chose this problem because both papers  (Han \& Gao, 2021a, 2021, Han \& Gao, 2021b, Kasai et al., 2018) tested their algorithms on this issue; thus, to facilitate comparisons between algorithms, we opted for the same problem.
>
> 3. Other Comments Or Suggestions:
>
> (1) Thank you very much for your suggestion. We will reduce the discussion on manifolds and exponential mappings in subsequent versions.
>
> (2) We apologize for any confusion caused. Due to space limitations, we provided only a simplified definition of IFO complexity in the submission. We will include a complete version and further explanations in subsequent papers.  The definition "For problem (1), an Incremental first-order oracle (IFO) takes an index $i \in \\{1,\ldots,n\\}$ and a point $x$, and returns the pair $\left(f_i(x), \operatorname{grad} f_i(x) \in \mathrm{T}_x \mathcal{M}\right)$. " and the explanations "The IFO complexity effectively captures the overall computational cost of a first-order Riemannian algorithm, as the evaluations of the objective function and gradient typically dominate the per-iteration computations."
>
> 3. We will modify $[n]$ to $\\{1,\ldots,n\\}$.
>
> 4. The first line of the formula above Equation (2) denotes the stochastic estimator for the Stochastic Recursive Momentum (STORM) algorithm, while the second line corresponds to the stochastic estimator for the Hybrid-SGD algorithm. The Hybrid-SGD estimator can be interpreted as a convex combination of the SARAH-type estimator and the stochastic gradient estimator. When $i_k = j_k$, the Hybrid-SGD reduces to STORM. And we propose a Riemannian extension of Hybrid-SGD, called R-Hybrid-SGD, as outlined in Equation (2). We will modify the formula above Equation 2 as follows:
>
> $v_k = \overbrace { (1 - \beta )( {v_{k - 1} - \nabla {f_{{i_k}}}({x_{k - 1}})} ) + \nabla {f_{{i_k}}}({x_k}) }^{STORM} \approx   \overbrace{(1 - \beta )(\underbrace {  {v_{k - 1} + \nabla {f_{{i_k}}}({x_{k }}) - \nabla {f_{{i_k}}}({x_{k - 1}})}}_{SARAH-type \\;estimator}) + \beta \nabla f\_{j_k} (x_k)}^{Hybrid-SGD}.$
>
> 5. Thank you for the reminder, we will modify it.

---

### Decision · Program_Chairs · 2025-05-01

**Decision:**

Accept (poster)

**Comment:**

All the reviewers are of the opinion that the paper has good merits. The unified viewpoint is interesting worth presenting to the community. In the final version, please take care of the minor corrections and include explanations.